# Submarine landslide megablocks show half of Anak Krakatau island failed on December 22nd, 2018

J. E. Hunt [1] [✉], D. R. Tappin [2,3], S. F. L. Watt [4], S. Susilohadi[5], A. Novellino [2], S. K. Ebmeier[6], M. Cassidy [7], S. L. Engwell[2], S. T. Grilli [8], M. Hanif [9], W. S. Priyanto [9], M. A. Clare[1], M. Abdurrachman[10] & U. Udrekh[11]

As demonstrated at Anak Krakatau on December 22nd, 2018, tsunamis generated by volcanic flank collapse are incompletely understood and can be devastating. Here, we present the first high-resolution characterisation of both subaerial and submarine components of the collapse. Combined Synthetic Aperture Radar data and aerial photographs reveal an extensive sub-aerial failure that bounds pre-event deformation and volcanic products. To the southwest of the volcano, bathymetric and seismic reflection data reveal a blocky landslide deposit ($0.214 \pm 0.036$ km$^3$) emplaced over 1.5 km into the adjacent basin. Our findings are consistent with *en-masse* lateral collapse with a volume $\geq 0.175$ km$^3$, resolving several ambiguities in previous reconstructions. Post-collapse eruptions produced an additional ~0.3 km$^3$ of tephra, burying the scar and landslide deposit. The event provides a model for lateral collapse scenarios at other arc-volcanic islands showing that rapid island growth can lead to large-scale failure and that even faster rebuilding can obscure pre-existing collapse.

[1] National Oceanography Centre, Southampton, UK. [2] British Geological Survey (BGS), Nottingham, UK. [3] University College London (UCL), London, UK. [4] School of Geography, Earth and Environmental Sciences, University of Birmingham, Birmingham, UK. [5] Marine Geological Institute, Bandung, Indonesia. [6] School of Earth and Environment, University of Leeds, Leeds, UK. [7] Department of Earth Sciences, University of Oxford, Oxford, UK. [8] Department of Ocean Engineering, University of Rhode Island (URI), Narragansett, RI, USA. [9] Research Center for Geotechnology, Indonesian Institute of Sciences (LIPI), Bandung, Indonesia. [10] Department of Geological Engineering, Institut Teknologi Bandung, Bandung, Indonesia. [11] Badan Pengkajian dan Penerapan Teknologi, PTRRB-TPSA, DKI Jakarta, Java, Jakarta, Indonesia. ✉email: James.Hunt@noc.ac.uk

Flank collapses at volcanic islands and their associated tsunamis are significant natural hazards[1–6], but the factors governing volcanic-edifice instability and landslide-emplacement remain incompletely understood. Detailed flank collapse reconstructions are key to understanding their mechanisms, timing and associated tsunami generation, which are essential to better identify the factors controlling failure and improving global monitoring of volcano instability hazards. Although characterising volcanic island flank collapses in the geological record (often 10 s kyr old) and numerically modelling their tsunamis have provided valuable insights into tsunamigenesis[7,8], there are uncertainties towards determining pre-event bathymetry, precise volumes, failure and emplacement processes, island geometry and tsunami magnitudes are all subject to uncertainties[4]. As a result, forecasting collapses and mitigating their associated cascading hazards are a major challenge[5]. The December 22nd, 2018 flank collapse and tsunami at Anak Krakatau (Fig. 1A), in the Sunda Strait, Indonesia is arguably the best-observed island-arc volcanic flank collapse. Although island-arc volcanoes are smaller than their counterparts at intraplate ocean-islands[2,3], they dominate the historical record of tsunamigenic flank collapses, and Anak Krakatau provides a valuable generally applicable analogue. Accurately determining parameters and emplacement of the 2018 Anak Krakatau flank collapse provides greater understanding of this event, and will enable improved numerical tsunami modelling, helping elucidate similar processes in contemporary island-arc settings.

The rapidly constructed Anak Krakatau volcano developed in the NE of the caldera basin formed by the 1883 Krakatau caldera eruption[9–12]. It emerged above sea level in 1927, rapidly growing to an eventual pre-collapse height of 333 m[9,12,13]. From June 2018, Strombolian volcanism built up the SW flank of the volcano with an estimated 54 million tons of lava and ejecta[5]. At 20:55:49[5] local time on December 22nd, 2018, following days of explosive eruptions the SW flank of the volcano collapsed, reducing the island height by over 50% and generating a tsunami (Supplementary 5)[5,13,14]. Within the caldera, on the adjacent islands of Sertung and Rakata, tsunami runup heights were over 80 m while on the coasts of Sumatra and Java they were up to 13 m[14–18]. The tsunami caused 437 fatalities, 14,059 injuries and displaced 33,719 people[19]. History shows, however, that landslide-tsunamis are not an isolated events, with comparable examples at Oshima-Oshima, Japan in 1741, Mount Unzen, Japan in 1792, and Ritter Island, Papua New Guinea in 1888, which also resulted in hundreds-to-thousands of fatalities[20–23].

The wealth of data now available on the 2018 Anak Krakatau event (Supplementary 2) provides a unique benchmark for testing models and hazard assessments of tsunamis from these volcanic island flank collapse[13]. Despite this array of observations, the collapse volume, the failure style and landslide headwall geometry all remain ambiguous[5,6,24–27]. Vigorous explosive volcanism followed the collapse, hindered satellite observations and obscured the failure plane by the rapid accumulation of post-collapse ejecta. The submarine components of the failure also remain undetermined. It is important to resolve these ambiguities to generate an accurate numerical model of the landslide for testing and improving numerical tsunami models, and to determine how and where the edifice failed, which can identify the controlling parameters of edifice failure and improve monitoring of flank instabilities. Submarine datasets hold the key to a more accurate understanding of the Anak Krakatau failure, since observations of the landslide deposit provide an independent means of measuring the failure volume and investigating landslide emplacement mechanisms. Here, we present comprehensive submarine observations of the event, and, for the first time, combine these with new subaerial imagery and satellite data that clarifies uncertainties

of the subaerial failure dimensions. The Anak Krakatau event, therefore, is a unique opportunity to fully parameterize a flank collapse, providing the necessary benchmark for accurately testing tsunami modelling solutions.

Previous tsunamigenic flank collapses (e.g. Ritter Island, 1888), lack accurate pre-collapse edifice and seafloor data, and have only limited tsunami observations, which restricts the resolution of numerical tsunami models[4]. The December 22nd, 2018 Anak Krakatau flank collapse and tsunami is the only major island-arc collapse event recorded by modern instrumentation, satellite technology, high-resolution seabed mapping and detailed observations of tsunami impact.

Here, we first provide new evidence of processes that preconditioned the 2018 failure. Using comprehensive, time series of satellite Synthetic Aperture Radar (SAR) images, field observations and aerial photographs we then define a new geometry of the subaerial component of the collapse scar. This is applied to a digital elevation model (DEM) of a pre-slide Anak Krakatau to provide a more accurate volume calculation of the subaerial landslide. We then present results from a new high-resolution seafloor and sub-seafloor hydroacoustic survey offshore Anak Krakatau, completed in August 2019, which provides a comprehensive view of the submarine landslide deposits. We compare these data to pre-event hydroacoustic data to resolve a total landslide volume, and insights into the submarine landslide emplacement dynamics.

## Results

**Failure preconditioning**. Sentinel-2 false-colour images of Anak Krakatau from June 2018 until the flank collapse show that lava flow accumulation during this period was restricted to the S flank (Fig. 1B; Supplementary 4 and 7)[5]. Strombolian ejecta were distributed equally on all flanks of the pyroclastic cone, but restricted to within the confines of the mapped failure area (Fig. 1C; Supplementary 4 and 7)[5]. The boundary between the pyroclastic cone and older structures from pre-1960 phreatomagmatic growth phase appears to have been a major control on the failure geometry. Landsat-8 and Sentinel-2 true-colour images of the crater at the summit of the pyroclastic cone between July and August 2018 show a reduction in crater size with a migration to the S, followed by enlargement towards NNW later in October–November 2018 (Fig. 1D; Supplementary 3 and 7). On 07/07/18 a NW-SE-striking fissure breached the surface on the N flank of the crater extending progressively SE as the cone built upward (Fig. 1E; Supplementary 3 and 7). Extensional faulting prior to collapse supports the previous determination of deformation of the entire SW sector of the island[5]. From 27/07/18, pale surficial deposits from fumarole activity are identified (Fig. 1E; Supplementary 3 and 7). These deposits align with the newly interpreted failure and imply active fluid flow and deformation over the eventual failure plane during the months before the flank collapse.

The location and extent of the 2018 failure was also potentially controlled by the underlying structure of the caldera margin, internal structural discontinuities, and location of previous instabilities. Near-annual topographic surveys by the former Geological Survey of Dutch East Indies, recorded an earlier landslide between 1941 and 1950 (Fig. 2A, B)[28–30] along a similar alignment, which written bulletins date to June 1949[31]. An aerial photograph in 1950 shows the extent of this failure (Fig. 2C), which resulted in a steep-sided, crescent-shaped, crater wall very similar to the 2019 landslide geometry (Fig. 2D). Past marine soundings also show that the volcano built atop and outwards from the steep 1883 caldera wall (Fig. 2E), likely creating gravitational instability.

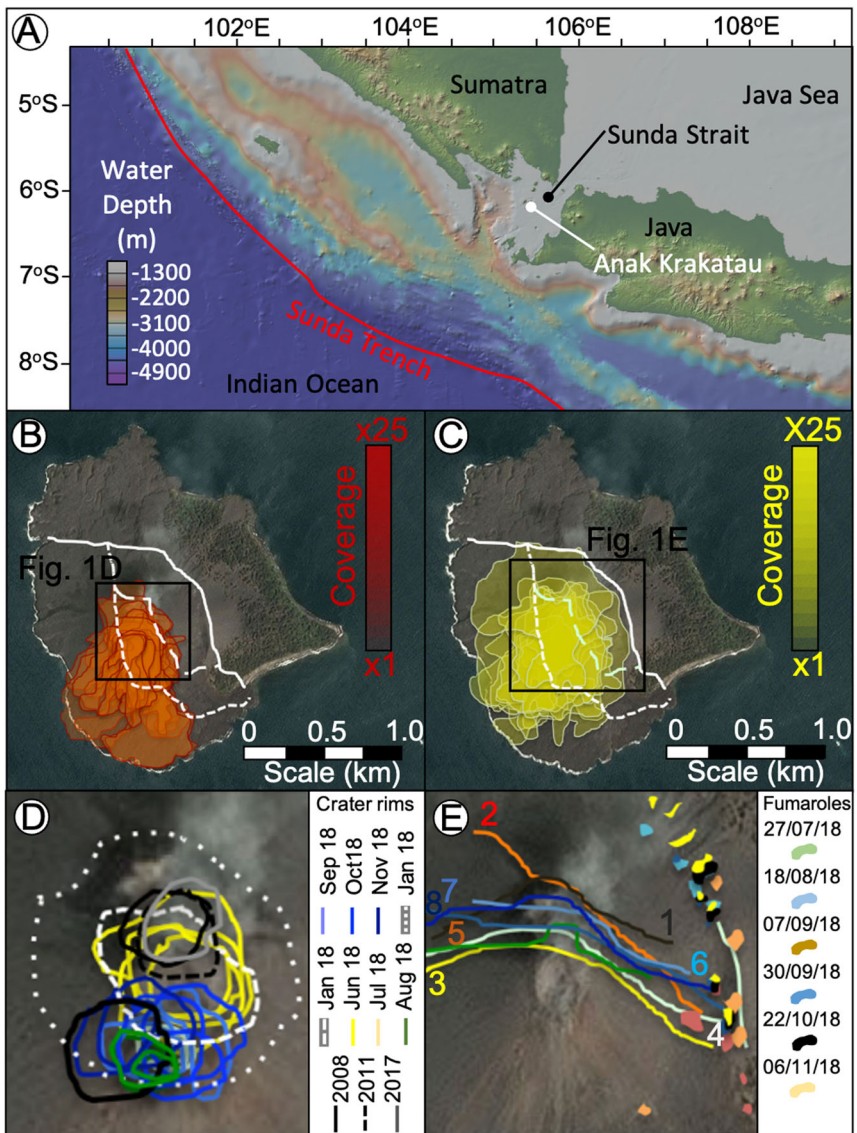

**Fig. 1 Maps of the study area location. A** Map of the Sunda Trench, showing the position of Anak Krakatau at an inflection point in the subduction of the Indo-Australian plate beneath Eurasia. **B** Sentinel-2 true-colour image (15/05/2018) of Anak Krakatau pre-collapse overlain with the distribution of effusive lavas between June 09/06/2018 and the flank collapse, based upon Sentinel-2 false-colour images[5] (Supplementary 4). New lavas exclusively accumulated on the SW flank within the limits of the new redefined failure plane. **C** Sentinel-2 true-colour image (15/05/2018) of Anak Krakatau pre-collapse overlain with distributions of Strombolian ejecta between 09/06/2018 and the flank collapse, based upon Sentinel-2 false-colour images[5]. Strombolian ejecta are restricted to within the limits of the failure plane. **D** Sentinel-2 true-colour image (15/05/2018) of Anak Krakatau pre-collapse overlain movement of the crater from 2008 until the flank collapse, based upon Landsat-8 and Sentinel-2 true-colour images[5] (Supplementary 3). The summit began as two vents in 2008, migrated to a northern location by 2011 where it remained until migrating southwards from July 2018. Location of (**D**) from inset on (**B**). **E** Sentinel-2 true-colour image (15/05/2018) of Anak Krakatau pre-collapse overlain with (i) position of faulting on the northern flank noticed from 07/07/2018 until the flank collapse and (ii) potential fumarole activity in this time, based upon Sentinel-2 true-colour images[5] (Supplementary 3). Numbered faults dated: (1) 07/07/18; (2) 09/07/18; (3) 01/08/18; (4) 18/08/8; (5) 12/10/18; (6) 06/11/18; (7) 16/11/18; and (8) 26/11/18. Faults appeared on the northern flank of Anak Krakatau from 07/07/2018 with fumarole deposits appearing from 27/07/2018 and aligning to the eventual trace of the failure plain. Location of (**E**) from inset on (**C**). **B–E** show the failure plane interpretations from previous published works (e.g. fine-dashed line, Williams et al., 2019; coarse-dashed line, Walter et al., 2019) and this study (solid line).

**The subaerial failure.** Volcanic plume and cloud cover prevented optical satellite observations of Anak Krakatau (e.g. Sentinel-2) during the period of flank collapse. However, a near-daily time series of constellation SAR images (Sentinel-1, ALOS-2, Terra-SAR-X, RADARSAT-2 and COSMO-SkyMed (CSK)) captures the major topographic changes at Anak Krakatau (Supplementary 9). These allow us to assess the position of the failure scar and aerial extent of the subaerial collapse, from which we determines the subaerial volume when applied to a 2018 DEM[13].

Combined high-resolution CSK images (Fig. 3B) with aerial photographs (Fig. 3C–F), further support our re-interpretation of the landslide scar (Fig. 3A, B).

The top of the collapse headwall on the high-resolution (pixel dimensions $0.3 \times 0.7$ m of range × azimuth, respectively) CSK SAR image from December 23rd, 2018, is a bright, curvilinear feature and is consistent with a $0.18 \, km^2$ subaerial failure area bounded by a NW-SE aligned scar (feature 5, Fig. 3B)[14]. This feature is also apparent in the lower spatial resolution Sentinel-1

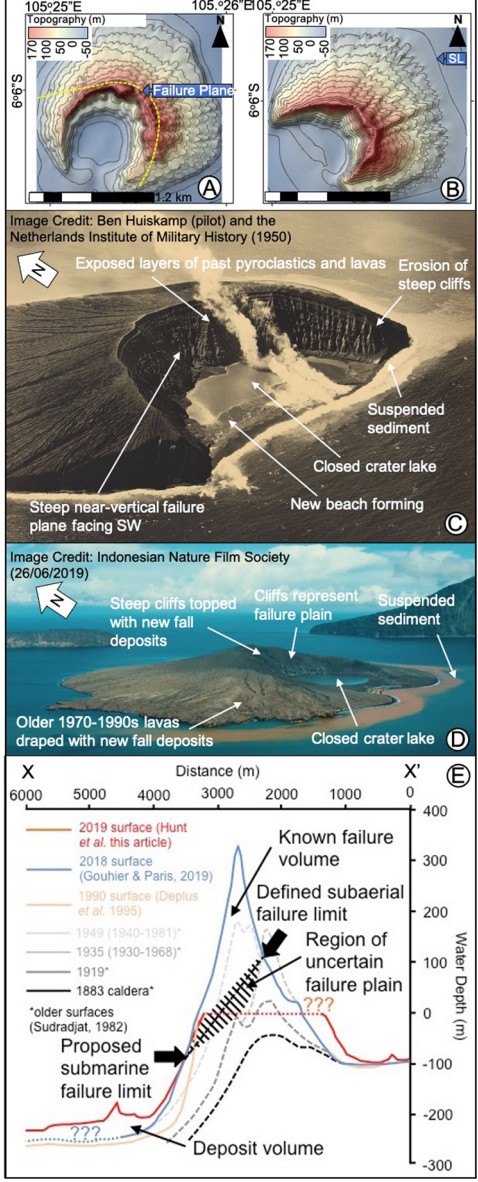

**Fig. 2 Evidence of past flank failures at Anak Krakatau and the influence of landsliding on the submarine slopes. A** Pre-collapse topographic contour map (10 m contours) of Anak Krakatau in 1941 (based upon Seibold & Seibold, 1996). The 1941 pre-collapse topography is overlain by the projected 1949 failure plane based upon the cliff top of the post-collapse island topography (Fig. 9B) and 1950 aerial photographs (Fig. 9C). **B** Topographic contour map (10 m contours) of Anak Krakatau in 1950 (based upon Decker and Hadikusumo, 1960). **C** Aerial photograph of Anak Krakatau taken in 1950 following a May 1949 collapse of the SW flank (Image credit: Militaire Luchtvaart and Royal Netherlands Air Force (1950) provided courtesy of pilot Ben Huiskamp); **D** Aerial still-image taken from drone footage of Anak Kraktau on 26/05/2019 after the 2018 flank collapse (Image credit: Indonesian Nature Film Society). The figure shows the similar failure geometry between collapses in 2018 and 1949, and the similar post-collapse recoveries. **E** Slope profiles of the SW flank of Anak Krakatau comparing our proposed shallow failure scenario with new 2019 post-collapse bathymetry, pre-collapse 1990 bathymetry and older slope profiles. These profiles show the recession in the slope at a water depth of −100 to −120 m resulting from the 2018 flank collapse. Failure aligns to the position of the position of the past Anak Krakatau tuff ring crater and the scarp of the 1883 caldera margin; suggesting the pre-existing caldera wall may provide a structure to influence flank failures.

SAR image from December 22nd, 2018 (feature 5, Fig. 3A) and later SAR images (Supplementary 9). This 2 km-long failure plane incorporates the active cone and followed the crater wall of the tuff cone constructed between 1927 and the 1960s (feature 4, Fig. 3B; Supplementary 7 and 9). Our interpretation is supported by post-collapse aerial photographs (Fig. 3C–F), which show where the pre-collapse shoreline has been cut by the failure plane. It is possible to identify this boundary precisely in photographs using the distinctive coastal lava deltas (features 1–4, Fig. 3A–F). We interpret a diffuse, sub-circular feature SW of the collapse scar as part of a co-eruptive plume similar to those observed in aerial photographs (feature 6, Fig. 3A–F). The co-eruptive plume likely had both subaerial and marine components and had diffuse boundaries and no clear internal features. Both ash-rich subaerial plumes and suspended volcaniclastic sediments at the ocean surface were well-documented during the eruption (Fig. 3C–F). The location of our defined collapse scar implies that the active volcanic conduit was cut beneath sea level, consistent with the observed transition to Surtseyan explosive activity immediately after the collapse due to magma-seawater interaction[14].

Comparison of the newly-mapped subaerial failure area with a pre-collapse 2018 DEM[13] gives a subaerial failure volume of $0.098 \pm 0.019$ km³. There is some uncertainty in the subaerial gradient of the failure plane, which from the CSK imagery is estimated at 30–40°, although up to 61° has been suggested[5]. The subaerial volume estimate is similar to other interpretations (e.g. 0.094 km³) that extend the failure plane east of the pre-collapse cone[13], but between two and twenty-two times larger than estimates where the failure plane limit is interpreted further to the west[25,29].

## Submarine landslide deposit architecture and dynamics

*Pre-collapse data—1990, 2017.* Bathymetry from 1990[11] has a lower (100 m) resolution than the 2019 post-event survey (5 m). From it we interpret a simple, steep-sided pre-collapse caldera basin, with the steepest slopes adjacent to the SW flank of Anak Krakatau. There are no landslide blocks on the seafloor SW of the volcano (Fig. 4A). The only features in the caldera floor are an eroded channel in the SW, an old block adjacent to Sertung Island and a probable, older, landslide deposit immediately N of Rakata Island. Across the pre-collapse caldera basin, seismic reflection profiles from 2017 support our interpretations of the 1990 bathymetry (Fig. 5A–C). Basin morphology is characterised by a faulted caldera basement infilled with 34–42 m of horizontally and regularly layered sediment, with a flat, near-featureless seafloor. The sedimentary fill is interpreted as pyroclastic deposits that accumulated between 1883 and 2017.

*Post-collapse data—2019.* By contrast, the 2019 bathymetry shows a deposit dominated by large, angular blocks that cover 7.2 km² of the caldera floor immediately SW of Anak Krakatau (Figs. 4B, C and 6A–C). These blocks are up to 1.5 km from the base of the SW flank (Fig. 6A–C) in 3–4 block 'trains', orientated N-to-S. Individual blocks are hundreds of meters in length (185–520 m) and width (300–500 m), and tower over 70 m above the surrounding seafloor (Figs. 4B, C, 6A–C). The angularity of the blocks and absence of gullying is consistent with their young age and lack of exposure to erosion. At the same time, the preservation of such large blocks suggests limited disaggregation and block interaction within the landslide mass during failure, translation and emplacement on the seafloor. The blocky landslide and intervening debris terminate at a well-defined ridge that rises 2–5 m above the current seafloor (Figs. 4B, C and 6A–C). There is no evidence of flank failures in the basin NE of Anak Krakatau.

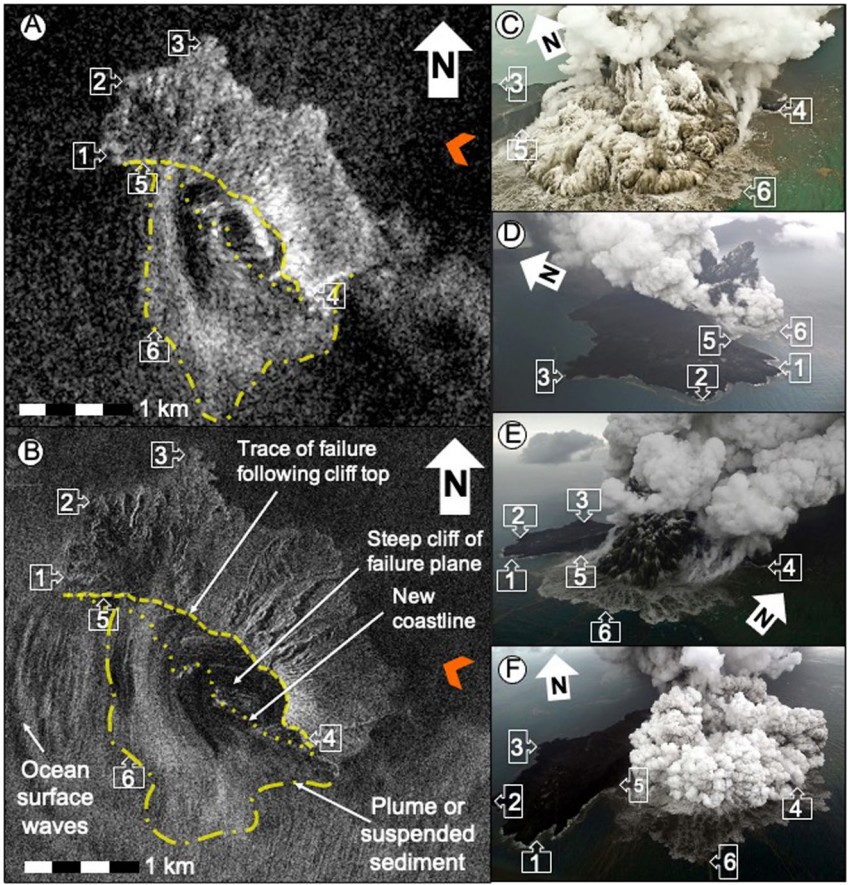

**Fig. 3 Satellite images showing the resultant landslide following the eruption shown in the photographs.** Comparison of Sentinel-1 SAR image of Anak Krakatau post-collapse on December 22nd, 2018 at 22.33 local time (**A**), with COSMO-SkyMed (CSK) SAR image of Anak Krakatau post-collapse on December 23rd, 2018 at 10.28 local time (**B**). Aerial photographs show Anak Krakatau erupting post collapse on morning of December 23rd, 2018. These photographs are used to validate the interpretations of the slide scar in the Sentinel-1 and CSK SAR images, and are used with permission of the photographers. **C** is credited to Didik Heriyanto taken at 17:01 on December 23rd, 2018; **D**–**F** are credited to Nurul Nidayat taken between 08:08 and 08:12 on December 23rd, 2018. Numbered markers are discrete tie points present in Fig. 2A–F. Yellow dashed line in Fig. 2A, B is the interpreted tip-line (cliff top) of the failure plane. Yellow dotted line in Fig. 2A, B is the interpreted cliff base (new coastline) where the failure plane projects underwater. Yellow dot-dashed line in Fig. 2A, B denotes the outline of the vertical plume emitted from the submarine vent following flank collapse. Orange arrows in Fig. 2A, B indicate the line of sight of the satellites. Publicly available Sentinel-1 SAR data is downloaded from the Copernicus Sentinel Hub portal. CSK SAR image is courtesy of the Committee on Earth Observation Satellite's Earth Observation (CEOS)'s Volcano Demonstrator.

Post-collapse seismic reflection profiles show the blocky morphology of the 2018 landslide deposits (Fig. 7A–C and 8A–C), which contrasts with the featureless caldera floor on the pre-collapse seismic profiles (Fig. 5A–C). Seismic reflection profiles SKC-01 to SKC-03 show a progressive decrease in block elevation, from up to 70 m in the NE near the Anak Krakatau scarp, to 20–30 m in the SW (Fig. 7A and C). There is internal layering within the blocks, with minor deformation except at their leading edges (Figs. 7A–C and 8A–C). Deformation at the leading edges of the blocks is represented by folding and (frontal) thrust faults, caused as the landslide mass experienced compression as it moved across and incised the seafloor (Fig. 7A–C). We suggest that the blocks represent fragments of the subaerial island and potentially parts of the submarine flank, and interpret the internal layering as interbedded pyroclastic deposits and lavas. Erosion of the seafloor (based on projections of the seafloor in adjacent undisturbed caldera sediments) is greatest beneath the largest blocks, as in line SKC-02 (e.g. 10–15 m, Fig. 7B, C), but generally <5–10 m (e.g. Figs. 7A and 8A–C). The incision reduces at the deposit margins with a stepped morphology at the contact

between the base of the landslide and the underlying strata (Fig. 7B, C).

On seismic reflection profiles SW of the leading block margin, two further units are identified that extend from the blocky area into the basin (Fig. 7A–C). The 8–10 m-thick lower unit is planar, with sub-parallel boundaries, tapering towards its SW periphery. There is no evidence that this unit eroded into underlying sediment (Fig. 7A–C). We interpret the lower unit as a debris flow that was contemporaneous with, or immediately followed, landslide block emplacement. It comprises a mix of chaotic and acoustically transparent facies, including rotated blocks individually 5–10 m in size (Figs. 7, 8; Supplementary 11). The debris flow deposit was likely generated from materials eroded from the seafloor during landslide emplacement. Within the vertical resolution of the available data, it is not possible to discern if a contemporaneous turbidite extends further into the basin, associated with the landslide and debris flow. The upper unit (7–14 m mean thickness, up to 25 m) is composed of low amplitude, parallel reflections. It overlies the lower unit SW of the landslide blocks but is also present between the blocks and onlaps

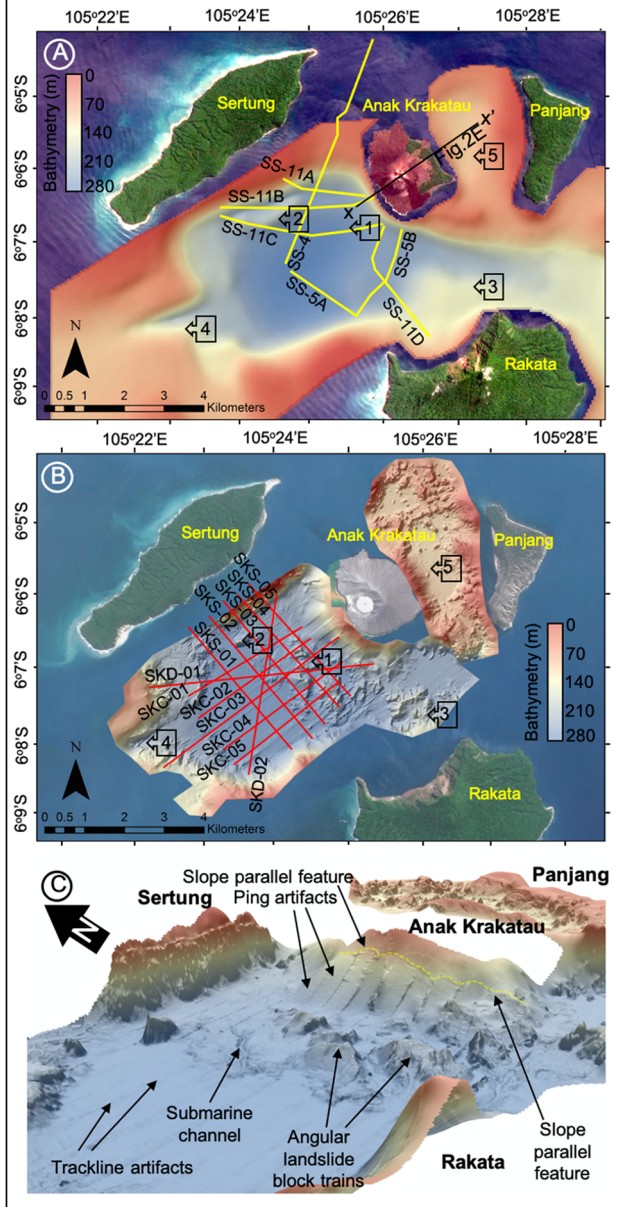

**Fig. 4 Pre- and post-collapse bathymetric maps of the seafloor surrounding Anak Krakatau. A** Raster surface of the pre-collapse bathymetry collected in 1990[9], showing the location of seismic reflection profiles collected pre-collapse in 2017. **B** Raster surface of the post-collapse bathymetry collected in 2019 showing the presence of a large landslide mass extending up to 1.5 km into the caldera basin. Also shown in Fig. 3B are the seismic reflection profiles collected post-collapse in 2019. **C** 3D rendering of 2019 bathymetry raster with hill-shade showing the large, blocky landslide deposit at the base of slope of Anak Krakatau's SW flank. Numbered arrows indicate similar features between the datasets: (1) base of SW flank, no landslide debris in 1990 (Fig. 3A), but a blocky landslide deposit in 2019 (Fig. 3B); (2) older block or intrusion adjacent to Sertung Island; (3) older landslide blocks N of Rakata Island showing burial and gullying from erosion; (4) erosional channel cutting the SW corner of the 1883 Krakatau caldera.

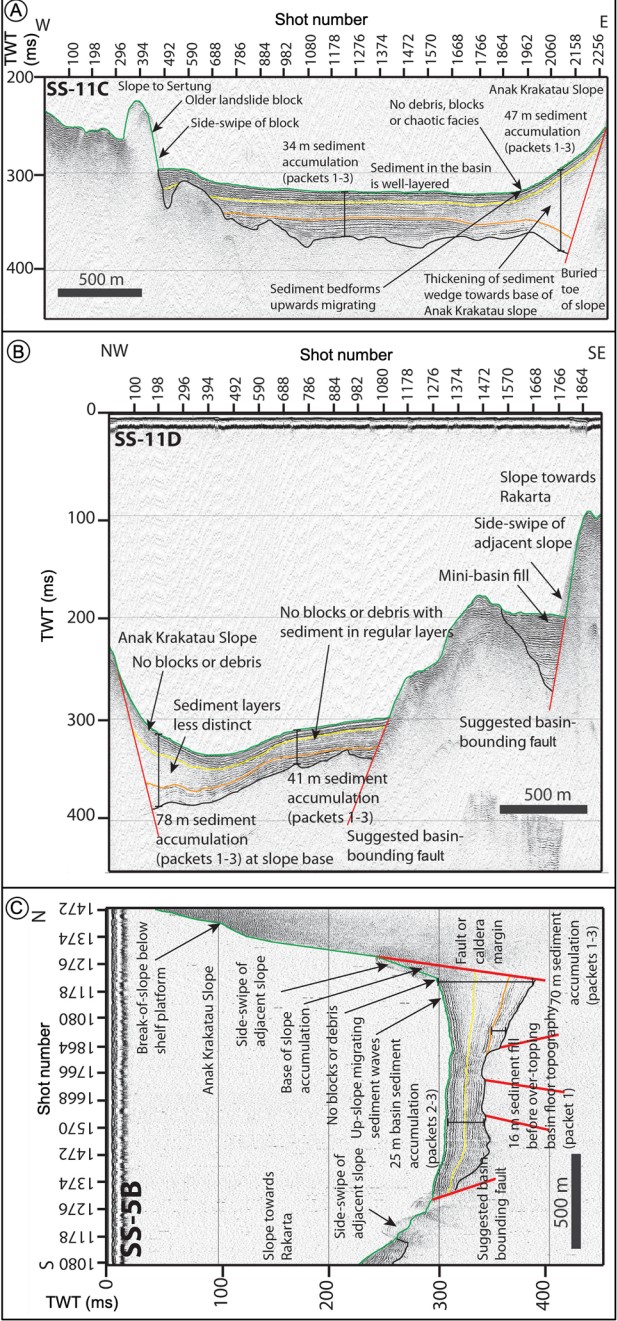

**Fig. 5 Seismic reflection SPARKER profiles across the Krakatau caldera basin collected in 2017, before the flank collapse, showing the simple basin geometry and lack of landslide debris at the surface.** Seismic reflection profiles SS-11C (**A**), SS-11D (**B**), and SS-5B (**C**) show 25–42 m (average 34 m) deposition between 1883 and 1990 within the caldera basin within three regular packages. There is no evidence of mass movements apart from minor, old, buried debris flows (represented by thin-bedded acoustically chaotic facies). Sediments likely from Anak Krakatau initially onlap basement caldera faults and infill irregularities in the basement topography. Basin stratigraphy above an initial sediment package is represented by horizontal planar reflectors.

onto the SW Anak Krakatau slope, indicating that this unit is the result of post-collapse sedimentation.

**Defining the submarine extent of the landslide scar.** Burial of the submarine failure plane means that its interpretation is

uncertain, but a minimum (shallow) and maximum (deeper-seated) failure scenario can be constrained (Supplementary 12) and compared with the independently derived deposit volume. Identifying the NE extent of the submarine landslide scar is

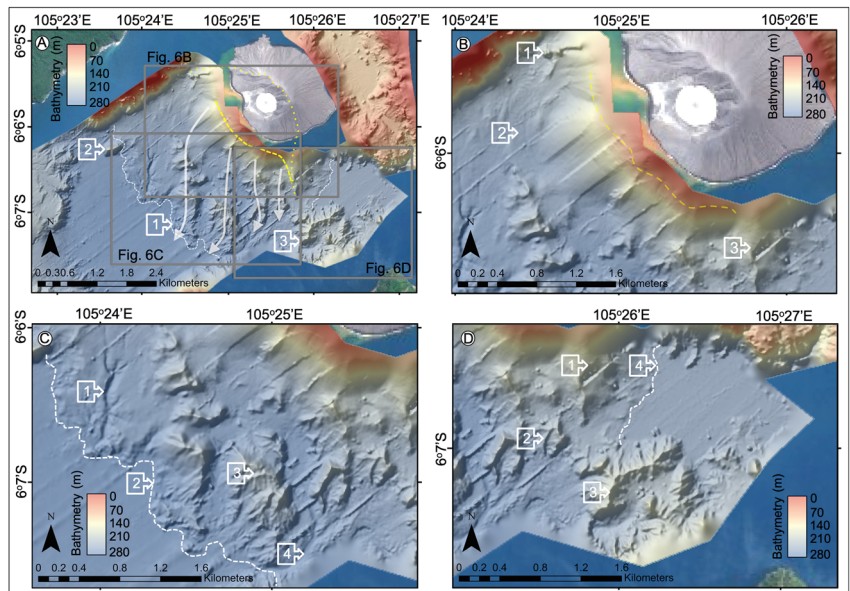

**Fig. 6 Post-collapse bathymetry of the seafloor surrounding Anak Krakatau collected in August 2019. A** Bathymetry of the SW and NE flanks of Anak Krakatau showing the clear evidence of a large landslide deposit in the caldera basin adjacent to the SW flank and no landslide materials in the shallow-water NE basin. **B** Magnified image of the bathymetry of the SW slope of Anak Krakatau transitioning into the landslide debris and blocks. **C** Magnified image of the bathymetry of the 2018 landslide blocks in the caldera basin adjacent to the SW flank of Anak Krakatau. **D** Magnified image of the bathymetry of the older identified landslide blocks immediately south of Anak Krakatau, which show greater degradation and gullying from exposure to submarine currents.

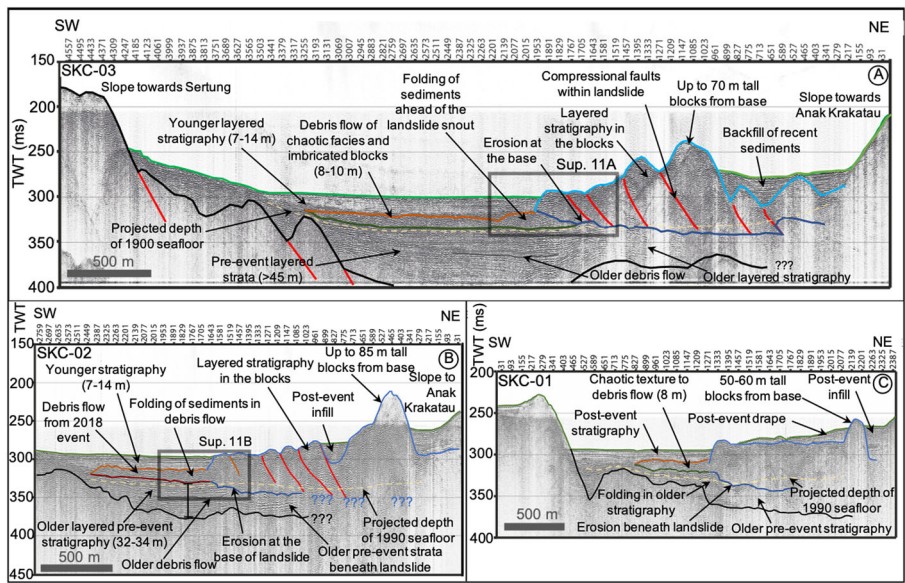

**Fig. 7 Downslope seismic reflection SPARKER profiles collected post-collapse in August 2019.** Profiles SKC-03 (**A**), SKC-02 (**B**) and SKC-01 (**C**) show the emplacement of a large landslide mass on the seafloor and its incision into the seafloor. Also shown is the presence of a secondary debris flow projecting into the basin from the toe of the landslide, which has been subsequently buried by excessive post-collapse sedimentation of eruptive materials. Also shown is the projected two-way travel-time-converted level of the 1990 seafloor (based on Deplus et al.[11] bathymetry) to demark the sedimentation in the basin since that time. The green horizon represents the seafloor; pale blue represents the top of the landslide blocks; dark blue represents the base of the landslide blocks; pale and dark brown represents the top and base of the buried debris flows, respectively; pale and dark grey represents the top and base of older buried debris flows; and red represents potential faults. Insets represent magnified images of the toe of the landslide and debris flow and are represented in Supplementary Fig. 11.

compromised by a data gap in the pre- and post-collapse marine surveys and rapid post-collapse deposition on the slope. However, on the SW submarine flank of Anak Krakatau, there is a recession in the 2019 slope at around −120 m water depth that cuts back by 20–50 m relative to the 1990 bathymetry (Fig. 2E). This recession in the slope corresponds with a subtle slope parallel feature at this

depth (Figs. 4C and 6B). This feature may represent the position where the 2018 failure plane cut the pre-collapse submarine flank, and the shallower slope gradient formed from subsequent infill. These observations allow us to define a minimum extent of the submarine failure surface. Alternatively, at around −160 m water depth there is a block-feature on the central slope (beyond the NE

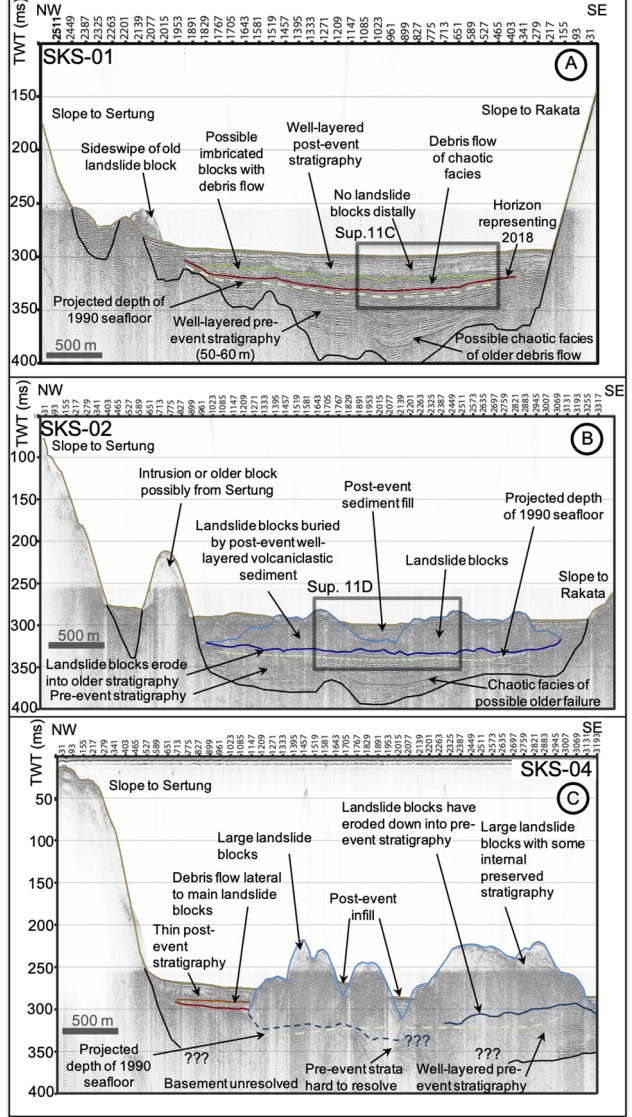

**Fig. 8 Slope-parallel seismic reflection SPARKER profiles collected post-collapse in August 2019.** Profiles SKS-01 (**A**), SKS-02 (**B**) and SKS-04 (**C**) also show the emplacement of a large landslide mass on the seafloor and its incision into the seafloor. Profile SKS-01 (**A**) shows the emplaced and buried debris flow found distally from the main landslide body. Profiles SKS-02 and SKS-04 (**B** and **C**, respectively) show the more proximal landslide deposits as large blocks that incise the seafloor, Also shown is the projected time-converted level of the 1990 seafloor (based on Deplus et al.[11] bathymetry) to demark the sedimentation in the basin since that time. All horizons are marked in the same way as Fig. 7. Insets represent magnified images of the debris flow and are represented in Supplementary Fig. 11.

limit of line SKC-03) and further breaks in slopes at ridges on the eastern and western lateral extents of the SW flank (Fig. 6B). These lateral ridges align with the offshore projection of the subaerial landslide scar and could be interpreted as sidewalls of a deeper and more laterally extensive failure plane.

**Landslide geometry and volumetrics.** From the available marine data we suggest two possible scenarios for the failure geometry of either: (1) a shallow failure propagating to a maximum −100 to −120 m water depth, which is limited to the depth of the intra-

island shelf and slope parallel features on the SW flank (Supplementary 12); or (2) a deep-seated failure propagating to the base of slope with a maximum water depth of −230 m and limited laterally to the ridges described above (Supplementary 12). Scenarios between these are possible, but have no basis in terms of any identifiable flank morphological features. Burial of the submarine failure plane means that its precise boundaries and basal gradient are uncertain.

For each geometry, the failure surface shape was further constrained by the subaerial headwall slope and by a requirement for the plane to cut the active vent at a depth of −25 m (estimated from the observation of Surtseyan activity[14]). Subtraction of the failure surface from a combined 1990 bathymetry and 2018 DEM topography yielded a shallow landslide volume of around 0.175 ± 0.015 km$^3$ (shallow failure scenario) and 0.313 ± 0.043 km$^3$ (deep-seated failure scenario) (Supplementary 12).

Our submarine datasets allow independent calculations of the landslide deposit volume, which can be compared with the scar volume estimates above. To do this, the pre-collapse 1990 bathymetry is first subtracted from the post-collapse 2019 bathymetry over the area of the proximal landslide blocks and debris (Fig. 9A). Individual blocks have volumes of 0.002–0.035 km$^3$, with a collective estimated volume of 0.064 km$^3$ (Fig. 9B).

A number of considerations and uncertainties influence the final deposit volume estimate and volume uncertainty (Table 1), which are detailed more extensively in the methods. Firstly, we account for differences in the resolutions of the 1990 and 2019 MBES data and the tidal and directionality artefacts in the 2019 MBES survey that cannot be fully corrected (Table 1). Applying accurate velocity modules in volcaniclastic sediments is challenging, thus uncertainty also arises during depth-conversion of the seismic reflection profiles[4] (Table 1). The initial deposit volume estimate is also subject to some uncertainties distinguishing the primary landslide mass from post-collapse sedimentation, pre-2018 sedimentation, and incorporated seafloor sediments (Table 1). The final volume must also account for marine sediment deposition between the 1990 baseline and December 2018 landslide. Another consideration countering the two uncertainties above is the possibility of erosion by the 2018 landslide extending deeper than the level of 1990 baseline seafloor. The 2019 seismic reflection profiles show up to 15 m erosion beneath the landslide blocks into pre-existing strata (Figs. 7, 8), which rapidly reduces to <5 m towards the landslide periphery, implying a total erosive volume of 0.025 ± 0.01 km$^3$ (Fig. 9C). To resolve these uncertainties, we apply several approaches (Table 1, methods). However, there are aspects of uncertainty that cannot be quantified where expert judgement is instead utilised, such as the trajectory of the failure plain beneath sea level.

Subtracting the post-collapse blocky deposit surface from the 1990 bathymetry gives a volume of 0.236 km$^3$ but taking account of the above uncertainties and based upon our initial estimate, we revise the volume of the landslide deposit to be 0.214 ± 0.036 km$^3$. We calculated (above) that 0.098 ± 0.019 km$^3$ of this volume was from the subaerial edifice. This means that the subaerial edifice contributed at least 45% of the landslide volume, with the remainder sourced from the submarine flank as well as bulking by incorporating seafloor material and expansion during transport.

In addition to the blocky landslide deposit, there is the debris flow to the SW (lower unit), buried beneath the post-collapse sediments observed on seismic reflection profiles (Figs. 7A–C and 8A). We interpolate the depths of the upper and lower boundaries of the debris flow deposit from the seismic reflection profiles and subtract them to provide a volume of 0.022 ± 0.006 km$^3$ (Fig. 9D). We suggest that the buried debris flow comprises eroded clasts of the pre-existing seafloor removed by the landslide blocks, rather

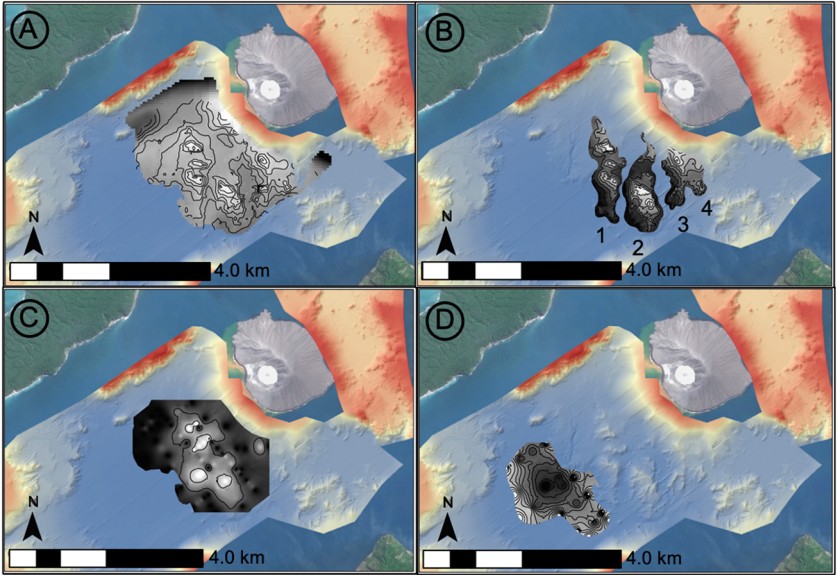

**Fig. 9 Landslide volumetric distributions. A** Difference map between 1990 bathymetry and new 2019 bathymetry over the landslide area; **B** Isolated difference maps of the four numbered landslide block trains (combined block trains 3 and 4); **C** Interpolated depth of erosion beneath the landslide block area based upon new 2019 seismic reflection profiles; and **D** Interpolated thickness of the buried debris flow beyond the proximal landslide blocks based upon new 2019 seismic reflection profiles.

than representing materials failed from the island flank. We also note that the estimated volume of the debris flow is very similar to our estimate of basal erosion by the blocky landslide. Therefore, the debris flow is excluded from our volume estimate of the primary landslide deposit.

**Past flank collapses**. On the 2019 bathymetry there is an older, blocky deposit N of Rakata Island not related to the recent landslide. The landslide blocks are similar in dimensions to those of 2018, but show extensive gullying, assumed to be from erosion by strong bottom currents, known to be present in the caldera. They contrast with the 'fresh' appearance of the 2018 landslide blocks (Fig. 5C). These blocks are not aligned to the failure direction from the SW flank of Anak Krakatau, so are more likely to be from N Rakata.

**Post-event deposition**. The acoustically transparent or chaotic 2018 debris flow deposit in the caldera basin (lower unit) is blanketed by a parallel-bedded (upper) unit, similar in seismic character to the pre-collapse caldera infill. We interpret this upper unit to be volcanic sediment generated between the failure in December 2018 and the survey in August 2019 by the vigorous post-collapse activity (Fig. 2C–F). However, we suggest this basin fill was more specifically generated in ~4-week period immediately following the collapse, when deposits from vigorous Surtseyan activity extensively modified the island's morphology during the main phase of post-collapse regrowth completed[31,32]. Although the 2018 flank collapse reduced the area of Anak Krakatau from 3.19 km[2] to 1.7 km[2], volcaniclastic deposits from the Surtseyan post-collapse eruptions led to the island exceeding its pre-collapse area within weeks, reaching 3.27 km[2 32,33]. An interpolated submarine thickness (average 14 m, from seismic reflection data) of post-collapse eruptive deposits across the caldera basin gives a deposit volume of $0.154 \pm 0.023$ km$^3$. Post-collapse submarine deposits must also have infilled much of the scar on Anak Krakatau's submarine flank, enabling the subaerial island to extend west of the collapse scar during the regrowth period, eventually enclosing the vent, and also resulting in the submarine failure scar being largely obscured, as described above.

Thus, from the volume of landslide deposit, less the subaerial collapse component, we estimate that the submarine scar volume (and thus the infilling sediment volume) is $\sim0.116 \pm 0.025$ km$^3$. In a previous study, the subaerial volume of post-collapse pyroclastic materials deposited on the island was calculated at $\sim0.029$ km$^3$ [13]. From these volumes, we estimate a present total post-collapse eruptive volume of $0.299 \pm 0.05$ km$^3$.

**Discussion**

Interpretations of satellite SAR backscatter for Anak Krakatau have implications for positioning the 2018 failure plane, determining the failure plane geometry, and calculating the subaerial landslide volume. During the event, eruption plumes (Fig. 2C–F) made discrimination between the sea-surface potentially rich in suspended debris, subaerial eruptive plumes, and the land, challenging on the 22/12/2018 Sentinel-1 SAR image (Fig. 2A). Later SAR images, from ALOS-2 (from 24/12/2018), Sentinel-1 (later from 25/12/2018), RADARSAT-2 (from 26/12/2018), TerraSAR-X (from 28/12/2018) (Supplementary 9), but most importantly CSK (from 23/12/2018; Fig. 2B), together with important aerial photography (afternoon 23/12/2018; Fig. 2C–F), allow accurately identification of the morphology of the southern part of the landslide scar for the first time.

From our two end-member failure plane geometries, the deep-seated scenario ($0.313 \pm 0.043$ km$^3$, Supplementary 12) implies a landslide volume greater than that measured in the deposit (although it is possible that some material remained within the collapse scar and is thus not included in our deposit calculations). From this, therefore, we consider the shallow failure scenario ($0.175 \pm 0.015$ km$^3$, Supplementary 12) is most likely, an interpretation supported by evidence from the bathymetry, although larger failure volumes cannot be fully excluded. This implies that the failure plane cut the seafloor on the slope at around $-100$ to $-120$ m water depth (Supplementary 12). This results in an initial volume close to 0.175 km$^3$ (subject to uncertainties in basal gradient and lateral boundaries), that bulked by sediment erosion and expansion during transport to form a deposit volume of $0.214 \pm 0.036$ km$^3$. Because elements of submarine failure geometry cannot be constrained, it is out deposit volume that

**Table 1 Uncertainties towards volume estimates.**

| Volume type | Aspect of volume | Details | Volume | Uncertainty | Volume uncertainty |
|---|---|---|---|---|---|
| Subaerial landslide volume | | Subaerial failure plane gradient from CSK imagery estimated 30–40°, maximum 61[5]. Failure plane gradient variations may account for uncertainty in volume around 10% of 0.098 km³. The base DEM has a 2 m per pixel resolution. The vertical accuracy of the January 2018 DEM was 3.7 m and the standard error at 2.2 m. The new DEM accounting for 2019 edifice growth provides additional uncertainty increasing from 10% up to 20%. | 0.098 km³ | 20% (up to 30%) | ±0.019 km³ (±0.028 km³) |
| Initial submarine landslide deposit volume estimate | Bathymetry resolution | The spatial resolution of the 1990 bathymetry is 100 m and 5 m in the 2019 bathymetry. The block S of Sertung is resolved in both surveys and is located in precisely the same location; while edges of the block in 1990 bathymetry are smoothed due to low resolution. Height of the same central location on the block is −182 m in gridded 1990 data and −163 m in the 2019 data; however, there is 14 ms TWT (11.9 ± 0.7 m) of drape in 2017 seismic reflection data (line SS-11C). This drape estimate is consistent with estimated 5–10 m of accumulation in the basin between 1990 and 2019. The block height in 2019 is closer to −175 m, implying the resolution difference in the data cause vertical errors of up to 10 m. Other comparative points of lower complexity on the basin margins indicate vertical differences of <5 m. | 0.236 km³ | ~20% conservatively (~15% realistically) | ±0.05 km³ (±0.036 km³) |
| | Bathymetry offsets in 2019 data | The full effects of tides and directionality of the 2019 bathymetric survey cannot be fully corrected between survey lines. Vertical differences between survey lines of up to 8.2 m are measured on the high gradient, proximal slopes, 1.9 m over the landslide blocks and 0.8 m over the basin. | | ~2.7% total (~4% over slope area ~2% over slide area) | ±0.013 km³ (±0.008 km³ on slope ±0.005 km³ over slide) |
| Landslide volume adjustment | Post-event sediment inclusion | Backfilled sediment accumulation landward of the landslide blocks on average 14 m over a slope area of 0.82 km². This accounts for 0.012 km³, with a likely range 0.008–0.016 km³. | 0.008–0.016 km³ | Up to 25% | ±0.004 km³ |
| | Pre-event sediment inclusion | The initial landslide deposit volume may include marine sediment deposition between the 1990 baseline and December 2018 landslide. Several methods are used to resolve that 0.035 ± 0.007 km³ of the initial volume may need to be removed:<br>1) Compare published low-resolution MBES surveys in 2016 (pre-collapse) and 2018[12] (post-collapse) to the 1990 pre-collapse multibeam bathymetry[9] and new 2019 high-resolution post-collapse multibeam bathymetry. Negligible-to-5 m accumulation implying up to 0.036 km³, but likely less, uncertainty from comparing bathymetries of different resolutions (Supplementary 12);<br>2) Compare the basin floor in 2017 from pre-collapse seismic reflection profiles to basin floor bathymetry in 1990 and 2019. Negligible-to-8 m accumulation implying up to 0.058 km³ volume (Figs. 7, 8), with uncertainty from velocity model used causing ±1m uncertainty;<br>3) Estimate sedimentation rate from pre-collapse 2017 seismic reflection data to project likely 1990–2018 sedimentation. This equates up to 8–10 m accumulation of 0.072 km³, with uncertainty from velocity model used causing ±1m uncertainty;<br>4) Compare two-way travel time-converted 1990 bathymetry to our 2017 and 2019 seismic reflection profiles (Figs. 7, 8). This equates <5 m accumulation of 0.036 km³, with uncertainty from velocity model used causing ±1m uncertainty. | 0.036–0.072 km³ | 1) 20% 2) -12% 3) -10% 4) 20% | 1) 0.007 km³ 2) 0.007 km³ 3) 0.007 km³ 4) 0.007 km³ |
| | | | 0.025 km | ~15% | 0.004 km³ |

**Table 1 (continued)**

| Volume type | Aspect of volume | Details | Volume | Uncertainty | Volume uncertainty |
|---|---|---|---|---|---|
| | Erosion into older strata | Erosion into the pre-collapse stratigraphy is up to 15 m beneath the largest landslide blocks, which decreases to <5 m towards periphery. This equates 0.025 km³ (Fig. 9C), with uncertainty based on velocity model used to covert seismic reflection data equating ±2 m uncertainty. Interpolation across the area presents an additional but unquantified uncertainty. | | | |
| Debris flow | | Interpolations have a potential error based upon the velocity model used to depth convert the seismic reflection data. This is estimated to be 1–2 m of the 5–10 m thickness used to calculate the 0.022 km³ volume. | 0.022 km³ | ~25% | 0.006 km³ |
| Post-event fill | | Tidal vertical offset in bathymetry survey lines is 0.8 m, equating a potential error of up to 0.004 km³. Uncertainty in a 1600–1800 m/s velocity model for depth converting surfaces from seismic reflection profiles provides uncertainty up to 2 m. | 0.023 km³ | ~15% | 0.004 km³ |

provides the best guide to the total collapse volume, with the lowest uncertainties.

Ye et al. (2019) calculated the landslide volume independently from seismic wave inversion methods to be up to 0.2 km³, which is within 10% of our calculation. Grilli et al. (2019) modelled a range of landslide volumes (0.22–0.3 km³) for the Anak Krakatau flank collapse and found that a 0.27 km³ source volume was most consistent with the recorded tsunami on the surrounding islands and coasts. Based on a finer-resolution grid simulation, these authors showed that the tsunami observations could also be explained by a shallower failure surface and a 0.22 km³ collapse volume[31]. Our new volume calculation is close to the lowest end of this volume range, similar to the authors' revised model. The landslide volume presented here is also supported by tsunami modelling estimates in other studies[24–27,34–37]; thus, indicating that our mapped landslide volume can explain the recorded tsunami without the need to invoke any additional source mechanisms.

The submarine and satellite observations illuminate the landslide emplacement processes and make the 2018 Anak Krakatau flank collapse dataset the most complete volcanic landslide-tsunami yet studied. Previous volcanic island flank collapses display a variety of mechanisms and emplacement dynamics. That of Ritter Island, 1888, resulted in a highly disintegrative mass, that transformed into debris flows running out for 70 km[4,38,39]. This 2.4–4.2 km³ volcanic collapse was highly erosive, with the distal deposits dominated by remobilized seafloor sediment[4,38,39]. Other volcanic island flank collapses, such as the 165 ka Icod landslide on Tenerife[2,3,40], disintegrate much less rapidly, forming block-rich deposits with strong elements of cohesive flow. Here, the collapse formed tongue-shaped debris lobes with large blocks rafted towards its periphery[2,38,39]. Alternatively, as at the 15 ka El Golfo landslide on El Hierro[2,41,42], even less block interaction and disintegration formed of randomly scattered large blocks spread across a broad, fan-shaped, debris apron. Finally, slides, such as Nu'uanu and Wailau on Oahu and Molokai islands, Hawaii, comprise much larger, kilometer-sized, blocks that did not disintegrate or scatter, but fractured and translated directly downslope, forming coherent block trains[43,44].

The 2018 Anak Krakatau landslide deposit is dominated by large (relative to the total collapse dimensions) angular blocks, distributed in distinctive block trains. These blocks travelled only 1.5 km into the basin with limited erosion (Figs. 3B, C and 5A, B), and yet it is clear that emplacement was sufficiently rapid to form an efficient tsunami source. An estimated slide duration[5] of 90 s implies emplacement velocities of at least 16.5 m s$^{-1}$; this is consistent with modelled slide velocities that range from 12 to 45 m s$^{-1}$ [26,45]. The Anak Krakatau deposit has morphological similarities with the (albeit much larger) Nu'uanu and Wailau landslides, with two important implications. First, that deposits are dominated by coherent, large blocks, with almost purely translational transport and relatively short travel distances into the adjacent basin, can originate via rapid *en-masse* failure and emplacement, and are an efficient tsunami mechanism. Second, that the style of fragmentation and transport may be difficult to predict and, at island-arc volcanoes, are just as varied as those observed in ocean-island settings, albeit on a smaller scale.

Here, we propose a dichotomy between translational slides, represented by the event at Anak Krakatau, and disintergrative and long-runout landslides, such as at Ritter in 1888. In many respects, the Anak Krakatau edifice is comparable to Ritter, being dominated by well-bedded mafic volcaniclastic deposits with interbedded lavas. At Ritter, however, this clastic failure mass appears to have been relatively weak and disintegrated rapidly, forming a highly mobile flow[35,45]. Although a superficially similar edifice, the pattern of disintegration at Anak Krakatau is in strong

contrast to Ritter, suggesting that the bulk properties, fragmentation and pre-collapse pattern of deformation preconditioning failure, may not be easily predictable. Nevertheless, for Anak Krakatau, assuming either a simple dense Newtonian fluid or homogeneous granular slide rheology, numerical models can reproduce the near- and far-field tsunami observations from our estimated failure volume[14,24–27,33–37,45,46]. While translational subaerial landslides are reported at other volcanic island sites, the detailed pre- and post-event surveys at Anak Krakatau and direct monitoring of the tsunami makes this an important benchmark event.

The Anak Krakatau landslide also resulted in an associated (now buried) debris flow (Figs. 7, 8), which we interpret as the result of the landslide blocks eroding the caldera floor. The debris flow, therefore, is a secondary effect of the original slide causing seafloor erosion or destabilization. Secondary failures are observed in collapse deposits offshore other volcanic islands, such as at El Hierro, Montserrat and Ritter[2,38,39,41,42,47]. While the Anak Krakatau debris flow unlikely contributed to the tsunami[48,49,50], it illustrates the potential for disruption of the seafloor beyond the termination of the primary landslide.

The collapse of Anak Krakatau occurred several months into a relatively intense phase of eruptive activity[5], and our observations suggest that this activity preconditioned failure by loading the SW flank with lavas and Strombolian ejecta (Fig. 1C). This area shows cumulative south-westward edifice deformation during this period[5]. As we also show, this deformation was accompanied by surficial fissures N of the crater 6 months prior to the failure (Fig. 1E), when the crater itself was migrating in the same direction (Fig. 1D). In addition, there was evidence of fluid flow above the location of the E boundary of the failure (Fig. 1E). Collectively, these observations suggest the incipient failure plane had at least partially formed during the months prior to collapse. The plane was also likely controlled by much longer-lived discontinuities within the edifice; the mapped failure is similar in geometry to a previous flank collapse in 1949 and follows the rim of the pre-1960 tuff cone. This suggests that the 2018 landslide scar location was controlled by the previous growth and failure history of Anak, which in turn was controlled by the underlying caldera basin wall (Fig. 2E).

We identify volcanism, fumaroles and faulting that preceded, and could have preconditioned, the flank collapse, which highlights processes that could be monitored at this and other islands to indicate potential future events. Our identification of the scale, geometry and emplacement of the 2018 Anak Krakatau flank collapse is important to understanding flank collapses at other volcanic islands, thereby underpinning improved numerical tsunami modelling[14,51]. We provide a well-validated landslide volume (0.214 ± 0.036 km$^3$) using high-resolution marine survey and satellite data. This is the first event volume estimated from the deposit, rather than from the projected failure plane, which is subject to greater uncertainties. Our landslide volume includes 0.098 km$^3$ (45%) from the subaerial failed edifice, and thus implies 0.116 km$^3$ (55%) is derived from the submarine flank. This favours a relatively shallow failure plane that cuts the submarine flank at −110 to −120 m water depth.

The landslide triggered a period of intense volcanism that generated a greater volume of material than was lost during the flank collapse, but that 90% of those materials were deposited offshore. Furthermore, the active modification of the seafloor during this time highlights that major volcanic environments can remain highly dynamic months after a flank collapse and illustrating the potential for collapse to initiate vigorous eruptive activity, which then rapidly obscures evidence of the collapse itself. This has implications for how easily we may identify, or not, evidence of prehistoric volcanic collapses in general.

The May 18th, 1980 Mt St Helens flank collapse and eruption resulted in a step-change understanding of explosive lateral blasts, edifice collapse and debris-avalanche emplacement processes[52]. The flank collapse of Anak Krakatau is the first recent volcanic island lateral-collapse tsunami observed by modern instrumentation, in particular, from a post-caldera collapse cone. It is also the first event to be studied by state-of-the-art, multiple disciplines, at high-resolution (temporal and spatial). It therefore provides another event, like Mt St Helens, that can form the basis for a similar step-change in understanding the mechanisms and hazards from rapid volcanic-island growth and destruction.

## Methods

**Analysis of past volcanic activity.** Sentinel-2 true and false-colour images were used to determine how volcanism from June to December 2018 may have preconditioned the SW flank of Anak Krakatau to fail[5,53,54]. True-colour (bands 4, 3 and 2) images showed the addition of lavas and outbuilding of the island, as well as the presence of steam venting and ash clouds (Supplementary 3) with the collapsed headwall already buried by new pyroclastic deposits on January 13th, 2019 so that details of the immediate post-collapse topography were no longer visible. False-colour images (bands 12, 11 and 4) also allow identification of hot surficial masses from background surfaces and can therefore show the addition of new lavas to the surface, whereby red-to-yellow-white colours represented new lavas of varying temperature (Supplementary 4). Repeated images of every Sentinel-2 pass of Anak Krakatau not obscured by cloud cover were mapped for the extent of new lavas. These maps were then overlain to produce a density map of volcanic activity in the pre-collapse period (Fig. 1C).

**Determining the geometry of the subaerial failure scar.** Radiometrically corrected and geocoded SAR data from multiple satellite platforms were interpreted visually to map the changes to Anak Krakatau during the December 22nd flank collapse. We constructed backscatter images from Sentinel-1 (S-1), ALOS-2, TerraSar-X and Cosmo-SkyMed (CSK) acquisitions spanning the collapse. We geocoded the images using a DTM clipped to the new island shoreline found from satellite imagery. SAR backscatter depends on (1) local slope angle relative to the satellite look direction, (2) roughness of the reflecting surface with respect to the incident radar wavelength and (3) dielectric constant of the surface (Arnold et al. 2018). It is therefore sensitive to both major changes in topography and surface cover. The CSK SAR image was acquired in Spotlight mode on December 23rd, 2018 (10:28 UTC) with a descending geometry at pixel spacing of 0.3 m in range direction and 0.7 m in azimuth direction (Fig. 3B). This image clearly shows a ~2 km long scar running NW to SE that reduced the area from 2.83 km$^2$ of the November 16th, 2018 to ~1.7 km$^2$ on the December 23rd, 2018[32].

The flank failure on December 22nd, 2018 left a steep cliff with a summit that has a high backscatter contrast associated with a change in local slope (Fig. 3A, B). The failure plane itself has numerous slope parallel features in backscatter; as these appear in both radar geometry we are confident that they are not geocoding artefacts. The coastline can be identified as an abrupt change in reflectivity, while eruptive plumes (subaerial and at the ocean surface) appear as more diffuse features that cross the coastline at some points (e.g. at the label for feature 5 on Fig. 3A).

The NW side of the collapse scar correlates with the part of the scar that we can see in the oblique aerial photos (Fig. 3C–F) and video from the afternoon of the 23rd (4 PM local time; therefore, several hours before this radar image). We interpret the arc-shaped backscatter features west of the vent as ocean surface wave surfaces moving away from the satellite look direction (~288 degrees). Ocean surface plumes of suspended volcaniclastic sediments seen in photographs correlate with diffuse features in the central part of the image.

Aerial photographs during the eruptions that followed the December 22nd, 2018 flank collapse on Anak Krakatau allow us to validate our interpretations of the SAR data by identifying distinctive physical features to act as tie points (Fig. 3A–C). These points include parts of the lava deltas formed in 1970–2000 on the N and SE of the island. We are then able to compare the curvilinear failure scar in the photographs and SAR images.

We constrain the slope angle of the subaerial failure plane from combining observations from satellite imagery and DTMs. The main scar surface has been reconstructed by creating a post-collapse DTM whose shorelines have been identified on the CSK image and whose elevations have been taken from the pre-collapse topography reconstructed in Gouhrier and Paris (2019). We estimate the slope of the collapse scar by interpolating between sea level at the new shoreline and the updated summit of the scar (max 150 m a.s.l.). The resulting sliding plane for the subaerial failure has a slope angle up to 60 degrees with median value of 35 degrees, comparable to alternative estimates (e.g. Walter et al.[5]).

**Bathymetry.** The pre-event bathymetry originates from a 1990 survey (Mentawai Cruise) of the Krakatau caldera basin by the R/V Baruna Jaya III (Supplementary 8A). A narrow-beam echosounder was deployed to map the seafloor. Navigation

for the data collection was recorded using uplinks to GPS and TRANSIT navigation satellites[9]. This data was previously compiled with past bathymetric data to generate a comprehensive map of the seafloor. Deplus et al.[11] applied a numerical model to combine bathymetric and topographic data of the Krakatau islands. Here, we georeferenced and digitised the bathymetric data from the 1900 survey[9]. The contours were then gridded at a 10 m resolution to provide a new pre-event bathymetric surface. This method has been applied in previous tsunami modeling studies to provide a pre-event bathymetric surface[12,48].

The post-event bathymetry of the Krakatau islands, including the Krakatau caldera basin was collected 12–19th August 2019 as part of a NERC-funded Urgency Grant (Supplementary 7B). Water levels and tidal measurements were recorded at a base station on Sebesi island (556793 E, 9343815 S, UTM Zone 48 S), which were used to calibrate both the bathymetric and seismic reflection survey data. A wooden vessel *Fortuna* was equipped with a *Teledyne* Reson T20-P multibeam echosounder (10–160° swaths) and *R2Sonic* 2026 bottom detection range finder (6 mm resolution). A *Teledyne* TSS-DMS 05 motion sensor was mounted on the multibeam echosounder transducers for post-processing corrections of vessel heave. Navigation was recorded from a *Trimble* SPS 462 GPS and data logger in a *Hydropro* data format. Track line artefacts were removed or smoothed in *CARIS* software. Digital data was analysed using *ESRI ArcGIS* software.

**Seismic reflection**. The same method for seismic reflection surveying was applied to both the pre-event 2017 and post-event 2019 surveys of the Krakatau caldera basin (Supplementary 9). The pre-event 2017 dataset was collected by a collaboration between the Indonesian Institute of Sciences and Marine Geological Research and Development Centre. The post-event 2019 dataset was collected by collaboration between the National Oceanography Centre, Southampton, the British Geological Survey, the Indonesian Institute of Sciences and Marine Geological Research and Development Centre.

The 2019 seismic reflection survey of the Anak Krakatau landslide deposits was completed from 20th August to 3rd September 2019. Here, a high-resolution Sparker methodology was applied. Acoustic pulses were generated by a multi-electrode Sparker using an *EG & G* power supply (model 230) and *EG & G* triggered capacitor bank (model 231). The system operated with a 1200 Kva power source and fired at a 500-millisecond rate with a 250-millisecond sweep rate. The system released energy in 400–600 Ws$^{-1}$ pulses with a frequency range of 300–10000 Hz. A neutrally buoyant streamer (*EG & G* eight-element hydrophone) was deployed 5 m from the acoustic source and 23–28 m behind the vessel. Signals received by the streamer were channeled to a *Khron Hite* 3700 band pass filter (30–3000 Hz) and then recorded using *Chesapeake* SonarWiz Shuttle with *SonarWiz* 4.0 recording software. SEG-Y data was analysed in *Petrel* software. The 2017 seismic reflection survey was collected under the same parameters but with a 360-millisecond sampling rate. The vertical resolution is <10 m, in the range 4–6 m with the working frequency towards the lower end of the range. The lateral resolution is 2 m based upon shots every 500 ms and a boat speed of no more than 4 knots. Lateral resolution away from the central track line suggests than features as great as 80 m away may influence the seismic reflection profile, potentially imaged as sideswipes.

**Volumetric analysis**. The new extent of the subaerial tip-line of the failure plane was applied to an existing pre-collapse 2018 DEM of Anak Krakatau to delineate the volume of the subaerial slide. The slope angle of the failure plane was limited by the new post-collapse coastline (slope angles 30–40°) and from calculations of the dip-angle of the shear failure (~60°)[5]. The 2018 DEM was combined with the existing 1990 bathymetric survey to provide a pre-collapse surface from which two subaerial-submarine geometries were modeled. These scenarios were limited laterally by ridges to the W and SE and then (1) slope features at −100 to −120 m, and (2) base of slope at −230 m. Failure surfaces were then contoured to provide appropriate concave spline fits between a steep and shallow end-member for each scenario.

With a pre-event (1990) and post-event (2019) bathymetric dataset it is possible to perform a volumetric analysis of the submarine landslide deposits in the caldera basin. There are resolution differences between the 1990 and 2018 bathymetric surveys that are considered (Table 1). The initial landslide deposit volume may include marine sediment deposition between the 1990 baseline and December 2018 landslide. Several methods are used to resolve this volume and the uncertainties are summarised in Table 1.

The landslide eroded down into the pre-collapse stratigraphy by up to 15 m beneath the largest landslide blocks and decreasing to <5 m towards the periphery. The level of erosion in seismic profiles is interpolated across the area of the slide. Interpolation across the area presents an additional but unquantified uncertainty. The seismic reflection surveys show that whilst the blocks have been erosive this is not aerially extensive and on an order of 10–15 m. Depth conversion of the seismic reflection profiles has uncertainty based upon the velocity model (Table 1).

**Seismic reflection depth conversion**. Thicknesses of landslide deposits and depths to particular horizons are estimated by converting two-way travel time (TWT) to depth. Modern deposits represented by those in the caldera basin,

especially volcaniclastic sediments, likely have varied consolidation and may have equally varied seismic P-wave velocities. Furthermore, the landslide blocks are intact and comprise older volcanic strata that may have much higher seismic P-wave velocities compared to unconsolidated sediments. Measurements of P-wave velocities in materials from the accretionary toe in the Nankai trough, which may be comparable to the compressive nature of deposits here range from 1600 to 2000 m/s[55,56]. Here, we use a conservative P-wave velocity of 1700 m/s attributed to unconsolidated but water-saturated sands. P-wave velocities of unconsolidated sands under varied vertical effective stress applied in a laboratory setting ranged from 1700 to 1800 m/s[56]. Thus, we used a P-wave velocity of 1700 m/s to depth convert seismic horizons to make estimates of deposit thickness and erosive depths. This is supported by common use of velocities of between 1600 and 1800 m/s to estimate the thicknesses of volcanic debris avalanches[5,37,55,56]. The challenge of applying an accurate velocity model in debris avalanches creates uncertainty summarised in Table 1.

To convert the 1990 bathymetry to TWT, in order to compare profiles of the bathymetry to the seismic reflection profiles required depth to be converted only using the P-wave of seawater between 1440 and 1570 m s$^{-1}$ (average 1500 m s$^{-1}$). This horizon was compared to seismic reflection profiles to estimate the amount of sedimentation between 1990 and 2018. This method was also applied to convert the TWT of the first reflector in the 2017 seismic reflection profiles to depth to provide estimates of pre-event basin depths.

**Historical island topographic analyses**. Topographic survey data was collated from publications of the Bulletin of the East Indian Volcanological Survey (Supplementary 6)[7,8] and regional reviews and report summaries[29,31,57]. These were georeferenced and digitised to generation new DEMs of Anak Krakatau for the available past surveys. Archived photographs were also collated for Anak Krakatau, including several from the National Gallery of Australia that show evidence of past mass wasting on the island. Topographic maps (Fig. 8A, B) show a potential landslide between 1941 and 1950. There is a report of a major recession of the SW crater wall in June 1949 (Supplementary 6), whereby, "The southwest wall of the crater was annihilated by wave-erosion and thus the crater lake has a crescent shape"[29]. While attributed to wave action at the time, the scale of the topographic change and reduced awareness of flank collapses in 1949 may mean that this event should have been attributed to a landslide. Photographs of Anak Krakatau in 1950 (Fig. 2C) show similarities to the post-landslide island in 2019 following the 2018 flank collapse (Fig. 2D), which provides additional support for this 1949 event to be attributed to flank collapse.

## Data availability

Figures 4–8 have accessible data that can be obtained in a raw format. Swath bathymetry and seismic reflection profiles obtained during the 2019 survey at Anak Krakatau will be made accessible to the public via the British Oceanographic Data Centre. These data will also be available upon request from the lead author J.E.H. from the time of publication. Swath bathymetry and seismic reflection profiles obtained during the 2019 survey at Anak Krakatau will be made accessible to the public via the British Oceanographic Data Centre. These data will also be available upon request from the lead author J.E.H. at the time of publication. Pre-event survey data from 1990 is already in the public domain, while pre-event seismic reflection data is held with co-authors S.S. and W.S.P. Photographs of the post-event eruptions have been published with permission of the photographers recognised. Additional photographs of Anak Krakatau in the supplementary information have been published with permission of the photographers recognised. Satellite data from COSMO-SkyMed is held privately; however, all other satellite data is publicly available. In particular, Sentinel-1 and −2 data can be accessed via Sentinel Hub.

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

## Acknowledgements

The authors thank the crew and technicians from *OC Enviro*, led by Ahmad Safii Maarif, for acquiring and processing the marine data. J.E.H. and M.A.C. recognise the National Environmental Research Council (NERC) Urgency Grant NE/T002034/1 "Failure mechanism and tsunamigenesis of the Anak Krakatau landslide on 22 December 2018". S.W., M.C., S.E. and M.A. acknowledge NERC Urgency Grant NE/T002026/1 "Interactions between eruptive and sector collapse at Anak Krakatau, Indonesia". D.R.T., S.D., S.E., A.N., S.G. and S.W. acknowledge NSFGEO-NERC Grant NE/S003509/1, "Tsunamis from large volume eruptions," that underpinned the Anak Krakatau research through its geographic focus on the Krakatau caldera. S.G. acknowledges NSF-GEO project GEO-17-56665 that has underpinned work on Krakatau and the Anak Krakatau landslide-tsunami. S.K.E. acknowledges a NERC Independent Research Fellowship and is associated to the NERC-BGS Centre for the Observation and Modelling of Earthquakes, Volcanoes and Tectonics (COMET). COSMO-SkyMed (C.S.K.) data have been provided by the Italian Space Agency through the Committee on Earth Observation Satellite's Earth Observation (CEOS)'s Volcano Demonstrator. Support from Institut Teknologi Bandung, Grant P3MI 2020 is acknowledged by M.A. that funded contributions to the marine and island surveys. The authors would like to acknowledge the funding to the Indonesian Institute of Sciences (LIPI) to complete the 2017 seismic survey. The authors would also like to thank: Professor Wahyoe Hantoro who led the 2017 seismic survey and made the data available for this study; the panel at Kementerian Riset dan Teknologi/Badan Riset dan Inovasi Nasional Republik Indonesia (RISTEK-BRIN) who advised on our proposal towards obtaining research permits; Dr. Willem Huiskamp and the Netherlands Institute of Military History for provision of the excellent photographs of Anak Krakatau in 1949 taken during his father's (Ben Huiskamp) flights as a military

pilot in Indonesia with the Royal Netherlands Air Force; and Dr. Raphael Paris for sharing topographic data. D.R.T., S.E. and A.N., publish with the permission of the CEO of the British Geological Survey (United Kingdom Research and Innovation). The authors would like to thank and acknowledge photographers Nurul Nidayat, Didik Heriyanto, Dicky Adam Sidiq, Oystein Lund Andersen, James Reyonds and the Indonesian Nature Film Society for provision of images for use within the study. The authors would also like to acknowledge the input and discussions with Professor Simon Day.

## Author contributions

J.E.H. was the lead author, developed the initial proposal, led the 2019 survey and analysed the marine survey data. D.R.T. helped develop the initial proposed, led the overarching field expedition to Krakatau in 2019 that enabled completion of the surveys at Anak Krakatau and contributed to the content and direction of the paper. S.F.L.W. led onshore field work at Anak Krakatau and the surrounding islands, helped analyse the marine data and contributed to the content and direction of the paper. S.S. helped present the original proposal to partners in Indonesia, provided 2017 pre-event survey data, and provided input to interpretations and paper construction. A.N. provided analysis of satellite data and inputs towards remote sensing interpretations in the paper. S.K.W. provided access to satellite data, interpretations of satellite data and input towards volcanological aspects of the paper. M.C. assisted in analysis of subaerial and submarine datasets and provided input towards the paper direction and content. S.L.E. provided data on historical development of Anak Krakatau, input towards interpreting remote sensed data and content towards the framing and discussions of the paper. S.T.G. provided valuable guidance on tsunami modelling for which information from the paper will be utilised and provided input towards the implications of study results. M.H. provided technical expertise towards acquisition fo the geophysical data and interpretations. W.S. P. provided expertise in processing of geophysical data, quality-controlled interpretations and input towards written results. M.A.C. helped construct the original proposal, provided assistance with interpreting marine survey data, volumetric analyses and input towards paper content. M.A. provided technical assistance and input towards methods. U.U. provided technical assistance towards acquisition and processing of field data.

## Competing interests

The authors declare no competing interests.
