## [Peer Review File · Nature Communications]

REVIEWER COMMENTS

Reviewer #1 (Remarks to the Author):

The article nicely summarizes the information available from the Anak Krakatau sector collapse, and adds important and novel seafloor data. This seafloor data is revealing the presence of large blocks that have their origin at the Anak Krakatau edifice. New volumetric constraints were elaborated and the presented information is of importance for a large readership.

The manuscript is mostly very well written, has some repetitive statements that can be easily deleted, otherwise arguments are clearly laid out, the figures are highly informative (although in places packed with unnecessary and unreadable information, while i missed more analysis on bathymetric data), and further supported by appendix materials (although here i missed the marine data in electronic form, geotiff or similar). I am recommending acceptance of the manuscript after the following general and minor corrections have been made.

General points:

1. Tone down criticism to earlier studies. I am not seeing the need to emphasize and repeat the lack and possibly wrong surface outline presented by earlier studies. A much more professional way of presenting your findings is to say that you can add further information to the southern sector that was hard to identify in earlier studies. So please tone down your paper bashing and repetitions, especially related to the SAR data, as i feel this rather weakens your own otherwise very nice results.
2. Figures are almost all unclear. This may be due to Covid19 related home office and a bad printer, but for most figures i had to go back to the electronic version to be able to read small font size, identify colors that are indistinguishable in greyscale and maybe not even needed.
3. Show more details on the bathymetry data. I am really suprised to see that earlier deposits are highlighted in close views, but the areas directly in line of the A.K. collapse are hardly identifyable. Please consider expanding the geomorphological analysis, include a marine slope or aspect map to better show the large blocks, analyse their geometry and sharp (or soft) limits more careful.

Other points with line numbers:

1. Reconsider the title, and mention the submarine blocks. Maybe something like "Deposition of submarine megablocks caused by the 2018 sector collapse of Anak Krakatau" or similar. What do the block implicate for the sliding dynamics?
2. Abstract highlights the SAR data, but it would be good to state that only by comparing high resolution CSK data and overflight photos you manage to reassess the southern amphitheater outline, which is larger than previously inferred.
3. L51: remove "largest and"
4. L52: why only in the past century? Ever?
5. L67: the exact timing was inferred from seismic and infrasound records in Walters et al. 2019
6. L79: maybe of interest here: at a small island named Kadovar, up to 5 landslide induced tsunami occurred in 2018. There was recently a paper published about it (<https://doi.org/10.1016/j.jvolgeores.2019.106704>).
7. L85: pls dont forget to mention the infrasound records
8. L91: probably the first (albeit not much recognized) paper describing the S1 and S2 data could be cited as well (Valade et al. 2019 <https://doi.org/10.3390/rs11131528>)
9. L97-98: please consider to release this data in the supplementary file. I absolutely do not understand why nowadays such data is not released together with the paper.
10. L113-125. To my understanding, previous workers agree to the amphitheater outline in the north, but uncertainty exists in the south. Here the present manuscripts adds further details. While the SAR data alone has to be taken with care (full of noise associated with the eruption plume and geometric distortions) i see the comparison to the photographs very convincing.
11. L149. How is the SAR data presented in Suppl 8 allowing to assess the volume? It allows identifying the failure scar (or better: the morphological scar), thus consider moving the reference up some words.

12. L153 and following. See my point 10 above. Also make sure to reduce repetitions.
13. L160. These aerial photographs are very nice, please consider including a close up of the photographs part with the morphological scar as it is really small in the printout.
14. L 174-75. Replace "incorrect" by "underestimated" is similar. Again, tone down criticisms and focus to present your novel results.
15. L177-179. I am a bit clueless how exactly the failure extent was used to constrain the subaerial failure volume with this precision. Please clarify method used and uncertainties.
16. L181. This volume calculations may be too simplified and need to be more critically assessed. As described by previous authors, accumulation of new materials was occurring very fast following the collapse, which is affecting simple volume estimations by DEM subtractions considerably. This is important to consider both subaerially as well as submarine.
17. L194. The featureless region might represent a deposition zone (see above point 16). I can not really see details in the bathymetry figure. Please consider expanding the bathymetry part of the manuscript.
18. L210. Please add details on the blocks, their dimension, shape, roundness, fault limits internal structure.
19. L214. you really mean "coherency" of the landslide? If a landslides desintegrates into blocks it is not maintaining coherency. Please justify or change your wording.
20. L223. In my print out this is the fifth figure.
21. L260. Please add details closer to the coastline at Anak Krakatau, especially where it collapsed. I do not understand why this part is not further detailed.
22. L280. The paper structure seems a bit strange here. Is it part of the discussion already?
23. L302-347. Reads like a discussion. Please consider highlighting novel results here, and move discussions down.
24. L384. Consider citing Giachetti et al 2012 here. They clearly had similar thoughts.
25. L434. The SAR data alone is not convincing, but together with the important aerial photographs the southern proportion of the morphological landslide scar can be identified for the first time. Please state it this way.
26. L449 and following. A lot of speculation on the failure plane. It might be valuable to compare to the failure planes (dip, strike, depth) as constrained by seismic data and by InSAR data (your reference 11).
27. These volumes slightly differ to other estimates by previous authors. I would find it very useful to include a table, either in main text or in appendix, that compares previous estimates and shows the new estimates made in this work.
28. L524. do authors consider rebuilding the island in the north too?
29. Acknowledgements. Data availability?
30. References. Please carefully check for correctness and completeness, add doi where possible, published volumes, issues and pages where missing. There are a number of other mistakes (e.g. you ref no 18) that I do not have to list here all.

Figure 1. I do not see any value of this figure. Everyone knows Krakatau, so figure a can be deleted, figure b is better seen in google earth, and figure c bashes on previous papers. Delete!

Figure 2. I find these earlier assessments very interesting. Note that a lot of the text is UNREADABLE in printout, such as scales, scale bars, and others. Make all text same font and size!

Figure 3. The SAR and aerial photographs are interesting, but authors must acknowledge that SAR alone is not conclusive in the southeastern part. Consider adding a close up of the scar in the subfigure "c" and "e" number 4.

Figure 4 and almost ALL other figures thereafter: Part of the text is unreadable in printout, such as scales, scale bars, and others. Make all text same font and size.

Figure 5. The bathymetry data could be further analysed. Identify the blocks, how large are they and what is their shape, elongation, height to width? Why there are closeups at the southwestern portion, but not from the proximal slope of the collapsed flank?

Figure 8. This figure is packed with stuff, part is not readable, and should be reduced in content. Consider moving panels a-c to the supplement, and keep D and E only. Panel E is interpretation loaded.

Figure 9 could be partly moved to the supplement.

Supplement: what is the point about the collapse calderas here? I do not see how it adds to the paper.

The sentinel data have been already published by several authors, and i am not sure how it supports your paper. I was annoyed to print out all of these without any information provided along with it.

Supplement 5: is it by chance that the reddish water color originates where authors draw the landslide amphitheater?

Supplement 7. Consider moving to the main body of the paper

Supplement 10. Where are the references here? All new and unpublished?

Reviewer #2 (Remarks to the Author):

The paper from Hunt et al. reports on the collapse of the Anak Krakatau volcanic Island, which occurred on December 22nd 2018.

The authors integrate satellite, bathymetric, and seismic data to quantify the total sediment volume that collapsed into the sea, providing for the first time a quantitative estimate of the subaerial and submarine components of the collapse. Compared to many other studies that investigated the 2018 Krakatau eruption, this is the first work presenting multibeam bathymetric data acquired soon after the event (which I think is the most relevant dataset) and higher resolution satellite images. Thanks to the new bathymetric data, the authors were able to quantify a total landslide volume of $0.214 \pm 0.025 \text{ km}^3$, of which $0.098 \pm 0.005 \text{ km}^3$ related to the subaerial failure. These values are overall similar to the results of previous studies, as Grilli et al. (2019) quantified a primary landslide volume in the range of $0.22\text{--}0.30 \text{ km}^3$, and Walter et al. (2019) and Gouhier and Paris (2019) quantified a subaerial failure volume of 0.102 km^3 and 0.938 km^3 , respectively. Other studies, as Williams et al. (2019) proposed a much smaller total volume. The authors interestingly highlight that the post-failure deposition during the regrowth of the volcano may obscure both the subaerial and submarine scars, thus affecting quantification of the total collapsed volume. The authors then present two possible scenarios for the failure geometry (with two failure planes at different depths), of which model 2 looks very similar to the model presented in Grilli et al. (2019).

The discussion starts with a comparison between the failed volume calculated in this study and the ones calculated in previous works. This comparison mainly points to the fact that the failure geometry of model 1, with a failure plane that cuts the seafloor at around -120 m water depth, is the most reasonable interpretation. Despite this conclusion can help a better understanding of the Krakatau event, it is not clear what this conclusion means in terms of a deeper understanding of the collapse processes that may occur in volcanic islands. The second part of the discussion mainly focuses on the morphology of the submarine landslides, and the authors suggest that the presence of blocks indicate a purely translational transport mechanism of relatively coherent masses downslope. Similar features have been observed in many other settings, modern and ancient, and interpreted in a similar way. Also this second part of the discussion does not provide a clear step forward in knowledge, apart for better describing the specific case study. The third part of the discussion focuses on the preconditioning factors, and the authors concluded that loading of the SW flank by lavas and volcanoclastic sediment coupled with topographic deformation preconditioned the failure. Walter et al. (2019) presented a similar conclusion. In summary, I found the discussion very descriptive, and the key statement presented in the introduction ("detailed reconstructions are key to advancing our broader understanding of tsunami generation from volcanic island flank collapses") is not further developed here.

Overall, the manuscript is well written, however, some parts can be improved. There are different statements that, despite logical, should be supported by additional data, as direct sediment

sampling of the seabed. See my comments below. The figures are clear, sometimes probably too dense. Some of the figures present errors and typos, one in particular (Figure 8) looks a like screen snapshot as you can see words highlighted by a red line and a selection box for the text. The captions also require some work, as information is missing. See my comments below.

An additional important point is the public availability of the data presented here. I did not find a statement on data availability, so I refer this concern back to the editor.

Considering the points above, I do not think this study introduces a major step forward in knowledge that is required by Nature Communications. This study presents a better quantification of the Krakatau event, but not a deeper understanding of the processes and products related to the collapse of volcanic islands that can be applied elsewhere. I think the authors does not present, and test, any new hypothesis that can help understanding such phenomena and evaluating the associated risk. For example: Why some volcanic collapses generate coherent blocks that move downslope for few kilometers, while others, as the 1888 Ritter Island event, resulted in a debris flow moving tens of kilometers away? How the knowledge gained in this field area can be used to predict the style of failure (and thus the risk to submarine infrastructure or the tsunamigenic potential) of volcanic islands in different settings? How such detailed calculation of the landslide volume affects the results of tsunami modelling? I think the authors should make an effort in improving the discussion, stating (and testing) one or more new hypotheses that will help understanding such kind of phenomena.

Additional Comments:

Lines 33-34: You use the word "extensive" twice. Would be better to specify the volume for both subaerial and submarine parts of the collapse?

Line 72: better to say here "runup heights".

Lines 78-79: Also the Mediterranean Sea, if you think about the Aeolian Islands or Santorini.

Lines 81-98: I understand there is a need to differentiate what have been already published from what is new in this study. I do not know if this list is the best way of doing it. Maybe what you have in lines 85-98 can be removed or discussed in the Methodology? I am saying this because the same information, presented in a better way, are reported in lines 113-140.

Lines 117-118: I would like to know here how much they differ. Reference 5 shows a volume very similar to what you present here.

Line 122: You haven't said why your approach reduces the ambiguity of previous interpretations.

Line 123: It is "new" for this study, or published by others?

Line 194: Why featureless? In line 196 you support this statement saying that a seismic line across the pre-collapse caldera shows a flat sea floor. I think that if a seismic line shows a flat sea floor you can not assume the sea floor is flat also outside the line track. It seems to me that the sea floor is featureless because of the low-resolution data. The block visible in figure 4B (line SS11-C) is not visible on the 1990 MBES because the lack of resolution. From the size of this block it seems also that the resolution of the MBES is much less than 100 m.

Lines 194-196: It would be good if you can point such features with arrows.

Line 201: How do you know that the deposits accumulated between 1883 and 2017? Provide more information in support of this statement.

Line 202: "<5 m thick", however in the methods you say "The vertical resolution is less than 10 m". It would be good the dominant frequency at such depths so to calculate the vertical resolution, as this information is not provided in the Methods.

Line 314: How do you know there is negligible to no drape? Do you have a sediment core?

Line 928: "This has been previously incorrectly interpreted as"... by who? And also, do you need this sentence in a figure caption? I think many figure captions present too much "interpretation" that is not needed.

Figure 2C,D,E,F: It would be good to have the time at which the photos were taken.

Figure 3C: Is there any way to improve the quality of the multibeam? The artefacts generated during the acquisition are quite large, and it seems also that the tide has not been correctly properly. This may affect the final volume calculation (see also figure 5B and 5C).

Figure 4: There is a typo "fault" instead of fault, repeated many times.

Figure 5C: It is not possible to read part of the text as it goes outside the figure.

Figures 6: There are errors in the caption, as the green horizon is not the only horizon at the sea floor, as also the black horizon towards SW and the blue horizon on top of the blocks are at the sea floor. I can not see the light grey horizon described in the caption, and there is no mention of the dashed brown horizon. I do not understand the Inset A, as the area in yellow is different from what has been interpreted in the main figure, same for Inset B.

Figure 7: "possible imbricated blocks with debris flow" is very hard to sell. In B, an intrusion should be accompanied by tilted reflections, while here it seems mainly straight onlaps.

Figurer 8: There is an error in labelling the figure (the first 8C should be 8B)

Figure 8C: No description of the different colors in the profile.

Figure 8D: It has to be redone for obvious reasons.

Figure 9A: The sea floor in orange should be in red, I think.

I do not know if figure 10 is necessary.

I hope the authors will find my comments helpful.

Reviewer #3 (Remarks to the Author):

Review of "Submarine observations show half of the island of Anak Krakatau failed on December 22nd 2018" – submitted to Nature Communications

This paper presents the result of a comprehensive survey of the geological and topographical changes due to the 2018 Anak Krakatoa landslide and eruption, combining multiple different sources of information, from SAR to results to a submarine investigation, among others. A comparison between the pre- and post-event situations is given emphasis. The main new contribution is the result from a comprehensive submarine field investigation, and the interpretation of the landslide volume, deposit, and composition.

There is no doubt that this is not only a comprehensive piece of work from the data collection point of view, it is also a thorough analysis attempting to combine the various datasets to interpret the release volume, deposits, and deformation as detailed as the data allow for. However, there is no attempt to model neither the flank collapse and the pyroclastic flows, nor the subsequent tsunami generation. Hence, the present paper mainly provides the relevant background and data interpretation that can allow for more accurate landslide and tsunami hindcasts of the present event in the future.

Many moderate and minor aspects must be clarified and improved before a publication can be considered. Some general comments are discussed first, followed by a series of line-by-line comments (partly elaborating and overlapping with the general comments).

The first general comment concerns the interpretation of the volume uncertainty. While there is no doubt that the authors' description of the data collection and comparison with previous datasets is trustworthy and thorough, there are elements of uncertainty in the interpretations that need better attention. In particular, there seems to be uncertainties (that are not necessarily quantifiable) related to:

- Less detailed mapping of the landslide near the volcano. Seismic lines as well as pre- and post-surveys are absent or more uncertainty is present here.
- Differences in pre-event bathymetries.
- Growth and changes in the volcano between the event and the survey.
- Sedimentation and debris transported away from the study area during and after the event.
- Erosion of the antecedent material.

All these items are discussed by the authors. However, in the quantitative analysis a rather narrow uncertainty bound is presented with little discussion about possible limitations. In particular, the first three bullet points represent sources of uncertainty that are not easily quantifiable. In fact, related to the third bullet point, it is highlighted by the authors in the abstract that "...shows how extremely rapid rebuilding can obscure evidence for such collapses". Moreover, the uncertainty in the slip plane configuration (figure 8C) shows that an essential piece of the submerged volume cannot be estimated by the data. This represents a major source of uncertainty that is not sufficiently discussed. I would suggest that when presenting the volume estimates and in particular the error estimates, the authors provide a more modest approach, and discuss these sources of uncertainty more elaborately. They should consider revising the description of the quantitative error estimates, or alternatively, skip them altogether. In summary, the quantitative analysis seems overly confident, and I suggest that the authors put more focus on describing uncertainties. This might also elucidate what the given error estimates really represent. It is necessary that this discussion is given in the main body of the paper, but the authors should consider also making more active use of the supplementary material for the quantitative volume estimates.

Related to the above comment, the authors describe several places relatively fine-grained features of scales 5-10 m (e.g. blocky structures). It is hard to convince oneself from the figures (for instance, Figure 6 & 7) that such small features (erosion, blocks) can be interpreted. To this end: What is the smallest spatial resolution in the seismic interpretation? How are these features interpreted? What is the basis for the interpretation? Consider also adding direct links to the supplement on how this interpretation is carried out. The authors should also consider providing more figures with close-ups to display the blocks better for the reader. Again, there must be more emphasis on the uncertainty in the interpretations.

Several relevant publications are not cited in the paper. In the line-by-line comments, some relevant additional papers are discussed. Some of them are more relevant than the ones listed by the authors to put the problem into context (e.g. Ramalho et al. and Paris et al., see below) and I was surprised not to see them cited if the intention is to understand better tsunami generation from volcano flank collapses.

Line-by-line comments

Line 39 - Is the 0.3 km³ in addition to the landslide volume? Can you say: "additional 0.3 km³ of tephra"?

Line 51 – add "arguably" before "best-observed"

Line 58 – It must be stressed that this paper, while thoroughly interpreting the geological footprint, does not actually attempt to reconstruct the dynamics of the event. Please add a sentence explaining this.

Line 78-79: There is a larger list of volcanic induced tsunamis for Indonesia (e.g. Ruang, Tambora tsunamis, possibly others), please review and cite Løvholt et al. (2012), Paris et al. (2014).

Line 102-103: Please revise the sentence starting with "This will provide...". Although this paper provides a rigorous mapping of deposits, failure volumes, etc., the landslide dynamics and the tsunami generation are neither analysed nor modelled.

Line 174: This is unclear. Which previous volume interpretations are you referring to? Please provide a reference and elaborate.

Line 179: The possible errors (uncertainties) provided throughout the document are very small. While the authors stress that interpretations are difficult due to dynamic changes in the volcanic growth, how can these volumes be interpreted with such confidence? I suggest that the authors discuss if there are other sources of uncertainty that may increase these uncertainty bounds. This is probably most relevant for the submerged part of the volume where uncertainties are presumably larger.

Line 235: How is the erosion quantified? The scales of 5-10 m are very small compared to the resolution provided in the figures. I cannot see from the seismic signals any evidence for such an accurate estimate of erosion. Please elaborate and illustrate how this can be interpreted in more detail. If this cannot be done, the description and conclusion should be revised as it might indicate a larger uncertainty related to erosion.

Line 247: This is related to the above comment: It is difficult to distinguish the 5-10 m size blocks with the present resolution (that this seems to be the limit of the seismic signal resolution). How were these rather small structures interpreted?

Line 302: You may also add that the geometry altogether, i.e. the distribution of landslide material and the run-out distance, is as important as the depth of the failure for interpreting the landslide dynamics.

Line 309: Elaborate and explain what the presented uncertainty bound is linked to (and equally important, what is not covered in the quantification).

Line 319: How are the backfill sediments interpreted? Please elaborate.

Line 344: How is this short 5 m scale resolved in the interpretation of the seismic signal? See also above comment, as this issue appears a few places.

Line 364: Add "buried" before "debris flow".

Line 402: Reading this sentence you get the impression that the volume is a surprise, although

several modelling studies have used this volume (and larger) in their analysis already. The estimate of the failure plane and volume even long before the event happened (Giachetti et al. 2012) was surprisingly close to the volume of the event. I suggest explaining this to set the work better into the present research context.

Line 440: Several other studies are now published and should be discussed. They have a broader investigation of the sensitivity of the landslide configuration and dynamics. Some of these are already cited by the authors, but not discussed here. Additional modelling work include Borrero et al. (2020), Ren et al. (2020), Zengaffinen et al. (2020). These analyses highlight new elements that the authors might want to discuss.

Line 454: What type of investigation to identify pre- and post-event data is conducted close to the landslide area / shoreline to estimate the landslide volume and identify deposits close to or on top of the failure plane on the rock slope? Is there an area close to the shoreline that is not covered, or more rudimentary covered? Could you please elaborate if there are data gaps or uncertainties related to the continuous slope changes due to the volcanic growth? Moreover, there are clear differences between the 1990 and 2017 datasets. How are the uncertainties in the interpretations taking these differences into account? The readers should be given more awareness about these uncertainties.

Line 461: Please rephrase "broader understanding of landslide dynamics and tsunami generation" to "a much more complete dataset and an interpretation that illuminate the emplacement process" or similar. The landslide dynamics and tsunami generation are after all not studied.

Line 466: Two essential works (Ramalho et al., 2015; Paris et al., 2017), where both tsunami deposit interpretation and joint modelling have been conducted, are not cited and discussed. They should be though, and they are the more relevant compared to the cited references if the authors want to discuss tsunami generation.

Line 477: Landslide modelling has evolved much since the developments in the 90's and there are now more advanced models available. More advanced granular models include $\mu(I)$ rheologies or other granular models, see e.g. Jop et al. (2006), Savage et al. (2014), or Si et al. (2018), to name some. These are more updated references valid for discussing future possibilities and should replace 11,35. However, I agree with the authors that the full complexity of the failure process cannot be captured by present models anyhow.

Line 490: Please stress that you are talking about the buried debris flow, not to be confused with the main landslide body.

Line 524-526: This may partly contradict the relatively narrow uncertainty bounds for the landslide volume. The authors should clarify better how the rapid changes in volcanic growth have been used in the interpretations.

Line 588: A. Paris is the first author in this paper (not R. Paris).

Line 973: Please include a legend that explains the colours of the different interpreted horizons such as the 1990 projected depth, etc.

References

Borrero, J.C., Solihuddin, T., Fritz, H.M. et al. (2020). Field Survey and Numerical Modelling of the December 22, 2018 Anak Krakatau Tsunami. *Pure Appl. Geophys.*
<https://doi.org/10.1007/s00024-020-02515-y>

Jop, P., Forterre, Y., & Pouliquen, O. (2006). A constitutive law for dense granular flows. *Nature*, 441(7094), 727-730.

Løvholt, F., Kühn, D., Bungum, H., Harbitz, C. B., & Glimsdal, S. (2012). Historical tsunamis and present tsunami hazard in eastern Indonesia and the southern Philippines. *Journal of Geophysical Research: Solid Earth*, 117(B9).

Paris, R., Switzer, A. D., Belousova, M., Belousov, A., Ontowirjo, B., Whelley, P. L., & Ulvrova, M. (2014). Volcanic tsunami: a review of source mechanisms, past events and hazards in Southeast Asia (Indonesia, Philippines, Papua New Guinea). *Natural Hazards*, 70(1), 447-470.

Paris, R., Bravo, J. J. C., González, M. E. M., Kelfoun, K., & Nauret, F. (2017). Explosive eruption, flank collapse and megatsunami at Tenerife ca. 170 ka. *Nature Communications*, 8, 15246.

Ramalho, R. S., Winckler, G., Madeira, J., Helffrich, G. R., Hipólito, A., Quartau, R., ... & Schaefer, J. M. (2015). Hazard potential of volcanic flank collapses raised by new megatsunami evidence. *Science advances*, 1(9), e1500456.

Ren, Z., Wang, Y., Wang, P., Hou, J., Gao, Y., & Zhao, L. (2020). Numerical study of the triggering mechanism of the 2018 Anak Krakatau tsunami: eruption or collapsed landslide? *Natural Hazards*, 102(1), 1-13.

S.B. Savage, M.H. Babaei, T. Dabros (2014), Modeling gravitational collapse of rectangular granular piles in air and water, *Mechanics Research Communications* 56, 1-10, <https://doi.org/10.1016/j.mechrescom.2013.11.001>.

Si, P., Shi, H., & Yu, X. (2018). Development of a mathematical model for submarine granular flows. *Physics of Fluids*, 30(8), 083302.

Zengaffinen, T., Løvholt, F., Pedersen, G. K., & Muhari, A. (2020). Modelling 2018 Anak Krakatoa flank collapse and tsunami—effect of landslide failure mechanism and dynamics on tsunami generation. *Pure Appl. Geophys* <https://link.springer.com/article/10.1007/s00024-020-02489-x>

**National
Oceanography Centre**

NATURAL ENVIRONMENT RESEARCH COUNCIL

National Oceanography Centre
European Way
Southampton SO14 3ZH
United Kingdom

Tel: +44 (0)23 8059 6666

<http://noc.ac.uk>

Direct line: 02380 596577

Email: James.Hunt@noc.ac.uk

30 October 2020

Re: resubmission of article NCOMMS-20-17603 – response to reviewer comments

Dear Reviewers,

Please find our responses to reviewer comments below.

We look forward to hearing your response,

Yours sincerely,

Dr James Hunt
Senior Researcher
Marine Geoscience Department
National Oceanography Centre, UK

FIGURE COMMENTS – ALL REVIEWERS

Author comments in **bold**. Here, Reviewers 1 and 2 provided a breakdown of each figure, while Reviewer 3 had comments regarding the data within the figures, rather than the figures themselves.

Figure 1 –

[Reviewer 1] I do not see any value of this figure. **We agree that this figure needs to be reimagined. It has been re-drafted to incorporate data that better provides detail on the preconditioning of the slide (panels B-E).** Everyone knows Krakatau, so figure a can be deleted **We disagree and other papers do show a map of the broader setting of Anak Krakatau,** figure b is better seen in google earth **We remove this panel,** and figure c bashes on previous papers **It was not our intention to ‘bash’ previous works. However, the miss-interpretations of those works need to be highlighted. The new figure only shows the outlines of the failure plane interpreted by Williams et al and Walters et al in comparison to ours and the new data we present. Delete!**

Figure 2 –

[Reviewer 1] I find these earlier assessments very interesting. Note that a lot of the text is UNREADABLE in printout, such as scales, scale bars, and others. Make all text same font and size! **Text should be Arial 14.** The SAR and aerial photographs are interesting, but authors must acknowledge that SAR alone is not conclusive in the southeastern part. **We believe that the SAR images alone would indeed conclusively show the extent of the SW failure, especially when considering all the available SAR images from the time (in Supplements, e.g. JAXA and RADARSAT images).** The field photographs (used here by permission of the photographers) help validate the interpretations. There is high backscatter from the water column that could lead to miss-interpretation of other authods. However, the CSK SAR image (B) is conclusive showing the top of the failure plane, where the failure plane interfaces with the sea surface, and the clear delineation of sediments in the water column. Consider adding a close up of the scar in the subfigure "c" and "e" number 4. **The photos were taken through airplane windows and magnifying regions of the photos is not revealing unfortunately.**

[Reviewer 2] Figure 2C,D,E,F: It would be good to have the time at which the photos were taken. **The times are provided in the caption to declutter the image.**

Figure 3 –

[Reviewer 1] *Text was pertinent for figure 2 not figure 3. This is rectified.*

[Reviewer 2] Is there any way to improve the quality of the multibeam? The artefacts generated during the acquisition are quite large, and it seems also that the tide has not been correctly properly. **The tidal affects and directionality of the survey create the offsets when the data from the combined tracklines is merged. Tidal corrections gave the present data. Further corrections served to smooth the data around the blocks too much. In the caption text will be provided to highlight the artefacts.** This may affect the final volume calculation (see also figure 5B and 5C). **Yes, this is possible and is dealt with in the new calculations. Please see methods section.**

Figure 4 –

[Reviewer 1] Part of the text is unreadable in printout, such as scales, scale bars, and others. Make all text same font and size. **The size is increased and all text should be the same font.**

[Reviewer 2] There is a typo “fault” instead of fault, repeated many times. **This is corrected.**

Figure 5 –

[Reviewer 1] Part of the text is unreadable in printout, such as scales, scale bars, and others. Make all text same font and size. The bathymetry data could be further analysed. Identify the blocks, how large are they and what is their shape, elongation, height to width? **We add more details about the blocks. We add a panel magnifying the slope-debris transition.** Why there are closeups at the southwestern portion, but not from the proximal slope of the collapsed flank? **This was because magnifications of that region gave little extra information. While we needed to rule out inclusion of the older blocks to the SE. A new panel of the desired region is added.**

[Reviewer 2] Figure 5C: It is not possible to read part of the text as it goes outside the figure. **This is amended.**

Figure 6 –

[Reviewer 1] Part of the text is unreadable in printout, such as scales, scale bars, and others. Make all text same font and size. **This should be rectified now.**

[Reviewer 2] There are errors in the caption, as the green horizon is not the only horizon at the sea floor, as also the black horizon towards SW and the blue horizon on top of the blocks are at the sea floor. I cannot see the light grey horizon described in the caption, and there is no mention of the dashed brown horizon. I do not understand the Inset A, as the area in yellow is different from what has been interpreted in the main figure, same for Inset B. **The insets are removed and placed within the supplementary information. The errors are rectified.**

Figure 7 –

[Reviewer 1] Part of the text is unreadable in printout, such as scales, scale bars, and others. Make all text same font and size. **This should be rectified now.**

[Reviewer 2] “possible imbricated blocks with debris flow” is very hard to sell. In B, an intrusion should be accompanied by tilted reflections, while here it seems mainly straight onlaps. **We agree, the ‘intrusion’ is a past landslide block, hence the onlaps. There would still be straight onlaps if it occurred shortly after the creation of the caldera basin before sedimentary infill began. However, we agree with the reviewer, it is most likely an older landslide block.**

Figure 8 –

[Reviewer 1] Part of the text is unreadable in printout, such as scales, scale bars, and others. Make all text same font and size. **This should be rectified now.**

This figure is packed with stuff, part is not readable, and should be reduced in content. Consider moving panels a-c to the supplement, and keep D and E only. Panel E is interpretation loaded.

[Reviewer 2] There is an error in labelling the figure (the first 8C should be 8B). Figure 8C: No description of the different colors in the profile. Figure 8D: It has to be redone for obvious reasons. **The figure has been redrafted.**

Figure 9 –

[Reviewer 1] Part of the text is unreadable in printout, such as scales, scale bars, and others. Make all text same font and size. **This should be rectified now.**

Could be partly moved to the supplement. **Agreed, part of the image have been placed in the supplements.**

[Reviewer 2] Figure 9A: The sea floor in orange should be in red, I think. **This is corrected. An unfortunate product of a colour blind lead author.**

Figure 10 –

[Reviewer 2] I do not know if figure 10 is necessary. **The figures have now been reworked. Parts of this figure have been merged with others.**

Supplements 1-2:

[Reviewer 1] what is the point about the collapse calderas here? I do not see how it adds to the paper. **This was to show the common occurrence of post-caldera cones in different settings. The map showing the distribution provides an anchor to discuss Anak Krakatau in a global context. The examples of post-caldera cones has been re-imagined to show a continuum of settings that either exacerbate or retard tsunami generation.**

Supplements 3-4:

[Reviewer 1] The sentinel data have been already published by several authors, and i am not sure how it supports your paper. I was annoyed to print out all of these without any information provided along with it. **These have been formatted onto fewer pages. We felt it important to provide the necessary information we used. These could be removed as the data is publicly available.**

Supplement 5 –

[Reviewer 1] is it by chance that the reddish water color originates where authors draw the landslide amphitheater? **This is where the ongoing volcanic activity is focused. The reddish water colour is suspended volcanoclastic sediment in the surface water and water column. It is what is giving high backscatter in the SAR images that was incorrectly interpreted as subaerial land by other others.**

Supplement 7 –

[Reviewer 1] Consider moving to the main body of the paper. **We felt that this is very important to support the methods and show the data coverage. However, we felt that alternate figures were better suited for the main body.**

Supplement 10 –

[Reviewer 1] Where are the references here? All new and unpublished? **The data does require referencing. However, these are new images of reprocessed data. The figure has been removed at this stage.**

TEXT REVIEWER COMMENTS

Reviewer #1 (Remarks to the Author):

The article nicely summarizes the information available from the Anak Krakatau sector collapse, and adds important and novel seafloor data. This seafloor data is revealing the presence of large blocks that have their origin at the Anak Krakatau edifice. New volumetric constraints were elaborated and the presented information is of importance for a large readership.

The manuscript is mostly very well written, has some repetitive statements that can be easily deleted, otherwise arguments are clearly laid out, the figures are highly informative (although in places packed with unnecessary and unreadable information, while i missed more analysis on bathymetric data), and further supported by appendix materials (although here i missed the marine data in electronic form, geotiff or similar). I am recommending acceptance of the manuscript after the following general and minor corrections have been made.

General points:

1. Tone down criticism to earlier studies. I am not seeing the need to emphasize and repeat the lack and possibly wrong surface outline presented by earlier studies. A much more professional way of presenting your findings is to say that you can add further information to the southern sector that was hard to identify in earlier studies. So please tone down your paper bashing and repetitions, especially related to the SAR data, as i feel this rather weakens your own otherwise very nice results. **We have toned the language appropriately now. While we do not feel that it was wrong to address other previously incorrect interpretations, we agree that the critique detracts from our own results. We do compare our results to other analyses, but feel now that the references are less repetitive and toned appropriately.**
2. Figures are almost all unclear. This may be due to Covid19 related home office and a bad printer, but for most figures i had to go back to the electronic version to be able to read small font size, identify colors that are indistinguishable in greyscale and maybe not even needed. **We have edited all the figures increasing the text size appropriately.**
3. Show more details on the bathymetry data. I am really suprised to see that earlier deposits are highlighted in close views, but the areas directly in line of the A.K. collapse are hardly identifiable. Please consider expanding the geomorphological analysis, include a marine slope or aspect map to better show the large blocks, analyse their geometry and sharp (or soft) limits more careful. **We have expanded the figures on the bathymetry whereby the magnified images better capture the slope and western limits of the blocks. We do provide measures of the blocks from lines 212 onwards. We can provide slope and aspects maps if these are deemed essential.**

Other points with line numbers:

1. Reconsider the title, and mention the submarine blocks. Maybe something like "Deposition of submarine megablocks caused by the 2018 sector collapse of Anak Krakatau" or similar. **We have reworked the title.** What do the block implicate for the sliding dynamics? **We cover this in the new discussions from lines 429.**
2. Abstract highlights the SAR data, but it would be good to state that only by comparing high resolution CSK data and overflight photos you manage to reassess the southern amphitheater outline, which is larger than previously inferred. **We make this distinction around line 30. We also make this distinction in text and in the methods additionally.**
3. L51: remove "largest and" **These lines are reworded.**
4. L52: why only in the past century? Ever? **We initially chose to look at those events that had better historical information and even instrumented data.**
5. L67: the exact timing was inferred from seismic and infrasound records in Walters et al. 2019 **This time of initiation is cited to Walters. The reference was there, but not positioned correctly, this is now rectified.**
6. L79: maybe of interest here: at a small island named Kadovar, up to 5 landslide induced tsunami occurred in 2018. There was recently a paper published about it (<https://doi.org/10.1016/j.jvolgeores.2019.106704>). **This is certainly interesting. The paper only presents satellite and photographic data, while here, we combine subaerial and submarine surveys of a flank collapse.**
7. L85: pls dont forget to mention the infrasound records **This is iterated in the comment on the timing of the generation around line 65.**

8. L91: probably the first (albeit not much recognized) paper describing the S1 and S2 data could be cited as well (Valade et al. 2019 <https://doi.org/10.3390/rs11131528>) **Yes this is an excellent paper, but we do not feel it is needed as other references contain pertinent information.**
9. L97-98: please consider to release this data in the supplementary file. I absolutely do not understand why nowadays such data is not released together with the paper. **There is intention to release the data for general public access through the British Oceanographic Data Centre (BODC), and the relevant links can be provided to the Journal when the data is uploaded. We can release the data to the reviewers under NDAs. At the moment the data has a moratorium period while we prepare publications. We are definitely open to sharing data, if fact it is a requirement from the funder, however, we have definitive routes for that until we publish.**
10. L113-125. To my understanding, previous workers agree to the amphitheater outline in the north, but uncertainty exists in the south. Here the present manuscripts adds further details. While the SAR data alone has to be taken with care (full of noise associated with the eruption plume and geometric distortions) i see the comparison to the photographs very convincing. **Taking all the SAR images as a time series our interpretations are supported. However, we definitively agree that linking the SAR images to the photographs provides additional validity to our interpretations. Arguably, neither dataset in isolation is definitive, but together the scale of the landslides is demonstrated.**
11. L149. How is the SAR data presented in Suppl 8 allowing to assess the volume? It allows identifying the failure scar (or better: the morphological scar), thus consider moving the reference up some words. **This section has been reworked. However, the SAR images help delineate the tip-line of the top of the failure scar and the outline of the scar at sea level. These outlines can define the area of the slide and can be applied to a DEM to provide the volume.**
12. L153 and following. See my point 10 above. Also make sure to reduce repetitions. **These sections have been reworked, and now hopefully reduce the repetitions.**
13. L160. These aerial photographs are very nice, please consider including a close up of the photographs part with the morphological scar as it is really small in the printout. **Unfortunately, even with the original photographs it is difficult to resolve the scar area when magnified because of the quality of the images. These images were taken through the obscured windows of the light-aircraft used to access the area. We can provide blow ups of the photographs in supplements if essential.**
14. L 174-75. Replace "incorrect" by "underestimated" is similar. Again, tone down criticisms and focus to present your novel results. **We remove this text and only present the results as recommended.**
15. L177-179. I am a bit clueless how exactly the failure extend was used to constrain the subaerial failure volume with this precision. Please clarify method used and uncertainties. **This should be clearer. The SAR images help delineate the tip-line of the top of the failure scar and the outline of the scar at sea level. These outlines can define the area of the slide and can be applied to a DEM to provide the volume. As the SAR images only resolve to sea level the volume calculation stops at 0 m, providing a subaerial volume. The table 1 provides additional information.**
16. L181. This volume calculations may be too simplified and need to be more critically assessed. As described by previous authors, accumulation of new materials was occurring very fast following the collapse, which is affecting simple volume estimations by DEM substractions considerably. This is important to consider both subaerially as well as submarine. **Yes. We did provide a lot of details originally on these calculations, and felt we did provide reference to the challenges in making these calculations. We have provided additional information now, and expanded on the uncertainties in table 1 and the methods.**
17. L194. The featureless region might represent a deposition zone (see above point 16). I can not really see details in the bathymetry figure. Please consider expanding the bathymetry part of the manuscript. **Yes, the landslide debris is deposited here and eroded by turbidity currents to generate the channels observed. We have provided additional images of the bathymetry.**
18. L210. Please add details on the blocks, their dimension, shape, roundness, fault limits internal structure. **These details were present. We provided summary ranges of their size. These details are provided in lines 212 onwards. There are also details in the bathymetry and seismic reflection profiles provided in the figures.**
19. L214. you really mean "coherency" of the landslide? If a landslides desintegrates into blocks it is not maintaining coherency. Please justify or change your wording. **This is reworded.**
20. L223. In my print out this is the fifth figure. **Figures are reworked and renumbered.**

21. L260. Please add details closer to the coastline at Anak Krakatau, especially where it collapsed. I do not understand why this part is not further detailed. **Additional details are shown in figure 6. The data on the slope is more heavily affected by tidal and directional survey artefacts.**
22. L280. The paper structure seems a bit strange here. Is is part of the discussion already? **We have reworked much of these sections.**
23. L302-347. Reads like a discussion. Please consider highlighting novel results here, and move discussions down. **We have attempted to rework much of this section. Summary of existing data is now in a supplementary table to focus attention on new data.**
24. L384. Consider citing Giachetti et al 2012 here. They clearly had similar thoughts. **Agreed.**
25. L434. The SAR data alone is not convincing, but together with the important aerial photographs the southern proportion of the morphological landslide scar can be identified for the first time. Please state it this way. **In the manuscript we tie the SAR images interpretations with the aerial photographs, and state the importance of using both together.**
26. L449 and following. A lot of speculation on the failure plane. It might be valuable to compare to the failure planes (dip, strike, depth) as constrained by seismic data and by InSAR data (your reference 11). **We do state the potential ranges in failure plane gradient from these different methods included now in table 1.**
27. These volumes slightly differ to other estimates by previous authors. I would find it very useful to include a table, either in maintext or in appendix, that compares previous estimates and shows the new estimates made in this work. **These information are provided in a supplementary table.**
28. L524. do authors consider rebuilding the island in the north too? **Yes. There is also some oversteepening of the slope of the N flank, but the water depths here are much smaller over a wider platform.**
29. Acknowledgements. Data availability? **Please see previous comment on data.**
30. References. Please carefully check for correctness and completeness, add doi where possible, published volumes, issues and pages where missing. There are a number of other mistakes (e.g. you ref no 18) that i do not have to list here all. **We have checked.**
-

Reviewer #2 (Remarks to the Author):

The paper from Hunt et al. reports on the collapse of the Anak Krakatau volcanic Island, which occurred on December 22nd 2018.

The authors integrate satellite, bathymetric, and seismic data to quantify the total sediment volume that collapsed into the sea, providing for the first time a quantitative estimate of the subaerial and submarine components of the collapse. Compared to many other studies that investigated the 2018 Krakatau eruption, this is the first work presenting multibeam bathymetric data acquired soon after the event (which I think is the most relevant dataset) and higher resolution satellite images. Thanks to the new bathymetric data, the authors were able to quantify a total landslide volume of $0.214 \pm 0.025 \text{ km}^3$, of which $0.098 \pm 0.005 \text{ km}^3$ related to the subaerial failure. These values are overall similar to the results of previous studies, as Grilli et al. (2019) quantified a primary landslide volume in the range of $0.22\text{--}0.30 \text{ km}^3$, and Walter et al. (2019) and Gouhier and Paris (2019) quantified a subaerial failure volume of 0.102 km^3 and 0.938 km^3 , respectively. Other studies, as Williams et al. (2019) proposed a much smaller total volume. The authors interestingly highlight that the post-failure deposition during the regrowth of the volcano may obscure both the subaerial and submarine scars, thus affecting quantification of the total collapsed volume. The authors then present two possible scenarios for the failure geometry (with two failure planes at different depths), of which model 2 looks very similar to the model presented in Grilli et al. (2019).

The discussion starts with a comparison between the failed volume calculated in this study and the ones calculated in previous works. This comparison mainly points to the fact that the failure geometry of model 1, with a failure plane that cuts the seafloor at around -120 m water depth, is the most reasonable interpretation. Despite this conclusion can help a better understanding of the Krakatau event, it is not clear what this conclusion means in terms of a deeper understanding of the collapse processes that may occur in volcanic islands. The second part of the discussion mainly focuses on the morphology of the submarine landslides, and the authors suggest that the presence of blocks indicate a purely translational transport mechanism of relatively coherent masses downslope. Similar features have been observed in many other settings, modern and ancient, and interpreted in a similar way. **The Anak Krakatau event has a known measured tsunami, thus not only does the discovery of a**

translational slide in this setting important, but it can be tied to the generation of this particular tsunami. However, Ritter experienced a flank collapse of a similar relative scale, but despite a similar composition the failure has highly disintegrative and erosive. Here, this translational slide only travelled 1.5 km into the basin, and maintained relatively intact. In comparison, the Ritter collapse generated an over 70 km run-out failure and debris flow. Also this second part of the discussion does not provide a clear step forward in knowledge, apart for better describing the specific case study. The third part of the discussion focuses on the preconditioning factors, and the authors concluded that loading of the SW flank by lavas and volcanoclastic sediment coupled with topographic deformation preconditioned the failure. Walter et al. (2019) presented a similar conclusion. **We now show additional information that show additional preconditioning factors. We show the brittle deformation and occurrence of surface fissures that support Walter et al. suggestion of SW deformation of the flank. We also show evidence of fluid flow in the form of fumarole activity showing active venting over the region that would be affected by the final failure.** In summary, I found the discussion very descriptive, and the key statement presented in the introduction (“detailed reconstructions are key to advancing our broader understanding of tsunami generation from volcanic island flank collapses”) is not further developed here.

Overall, the manuscript is well written, however, some parts can be improved. There are different statements that, despite logical, should be supported by additional data, as direct sediment sampling of the seabed. See my comments below. The figures are clear, sometimes probably too dense. Some of the figures present errors and typos, one in particular (Figure 8) looks a like screen snapshot as you can see words highlighted by a red line and a selection box for the text. The captions also require some work, as information is missing. See my comments below.

An additional important point is the public availability of the data presented here. I did not find a statement on data availability, so I refer this concern back to the editor. **The data will be made publicly available via the BODC. At the moment there is a moratorium with our funder to allow us to publish the results before the data is made public. We can provide the data to the reviewers upon request and with appropriate NDAs with the National Oceanography Centre.**

Considering the points above, I do not think this study introduces a major step forward in knowledge that is required by Nature Communications. This study presents a better quantification of the Krakatau event, but not a deeper understanding of the processes and products related to the collapse of volcanic islands that can be applied elsewhere. I think the authors does not present, and test, any new hypothesis that can help understanding such phenomena and evaluating the associated risk. For example: Why some volcanic collapses generate coherent blocks that move downslope for few kilometers, while others, as the 1888 Ritter Island event, resulted in a debris flow moving tens of kilometers away? **This comparison to Ritter is valid, and highlights the importance of the study. A more disintegrative and long run-out slide was expected here, at Anak Krakatau. Another key aspect that does show a step change is the huge volume of post-event materials produced, and that the majority of these volcanoclastic sediments are deposited offshore. Thus deposit the rapid recovery of the island post-event, this rapid recovery actually belies the true scale of recovery and the amount of volcanic debris produced. However, the major novel feature of this study is the combination of subaerial and submarine investigations of the slide so soon after the event.**

We have outlined the novelty and importance of our study as requested. Our study documents, for the first time, direct and complete high-resolution observations of the landslide deposits from the Anak Krakatau flank collapse on December 22nd, 2018. Firstly, there is the novelty of the completion of the submarine studies of this contemporary volcanic island flank collapse so soon after event, at very high resolution, and with pre-event data for comparison. However, this is not the only novelty, but does warrant highlighting and emphasising. It is important to stress the novelty of this opportunity and the data we have brought together for this study of both the subaerial and submarine components of the flank collapse at Anak Krakatau. Secondly, the Anak Krakatau event has given us greater insights into the failure and emplacement processes in these settings. The Anak Krakatau event has shown that landslides in these settings can remain predominantly intact and translate only a limited distance into the basin. In contrast, similar destructive events at similar volcanic islands are more disintegrative, more erosive and travel further into the basin (such as Ritter, as suggested by reviewers). Thus, this study shows

the dichotomy of potential slides in these volcanic settings capable of generating destructive tsunamis. Thirdly, we show the excessive additional generation of tephra from the co-eruptive events at the time of the flank collapse, and that these materials are deposited in the submarine realm on the slope and caldera basin. Single pyroclastic flows entering the submarine realm, such as at Montserrat, have been studied but have demonstrated that these deposits from extreme volcanic events are often sub-metre in thickness. Meanwhile eruptive sequences tens of metres thick often represent thousands of years of sediment accumulation. Here, at Anak Krakatau our results show an unprecedented thickness of co-eruptive and immediately post-event volcanoclastic accumulation from less explosive Surtseyan activity. We show that the new volume estimate from these eruptions is greater than the volume lost from the island flanks during the flank collapse.

How the knowledge gained in this field area can be used to predict the style of failure (and thus the risk to submarine infrastructure or the tsunamigenic potential) of volcanic islands in different settings? **The study shows the flaws in predicting the failure extent and style, especially without all the information. For example, the information of fissures and fumarole venting in this study show the SW flank deformation in response to loading from the lavas and Strombolian ejecta, while the venting occurred along the E and S extent of the failure. Knowledge of the past slides, as shown here, also defined a consistent weakness defining the N extent of the failure. Thus will this information, it may have been possible to predict the size of the subaerial failure.** How such detailed calculation of the landslide volume affects the results of tsunami modelling? **Producing tsunami models is a feature of follow on work. Completing the work here would make the study excessively long and not suitable for this format. The tsunami modelling warrants a focused study.** I think the authors should make an effort in improving the discussion, stating (and testing) one or more new hypotheses that will help understanding such kind of phenomena. **We have reframed the paper to highlight the novel science being presented.**

Additional Comments:

Lines 33-34: You use the word "extensive" twice. Would be better to specify the volume for both subaerial and submarine parts of the collapse? **This has been edited.**

Line 72: better to say here "runup heights". **This is corrected.**

Lines 78-79: Also the Mediterranean Sea, if you think about the Aeolian Islands or Santorini. **We wanted to reference recent historic events. Certainly, Santorini is a suitable**

Lines 81-98: I understand there is a need to differentiate what have been already published from what is new in this study. I do not know if this list is the best way of doing it. Maybe what you have in lines 85-98 can be removed or discussed in the Methodology? I am saying this because the same information, presented in a better way, are reported in lines 113-140. **This section of the paper has been reworked, and addresses this repetition.**

Lines 117-118: I would like to know here how much they differ. Reference 5 shows a volume very similar to what you present here. **We agree that there have been similar estimates. However, the post event DEM from Walter et al. is taken after the island has already significantly recovered. The 0.102 km³ volume estimate from Walter et al. is greater than our estimate, but within error.**

Line 122: You haven't said why your approach reduces the ambiguity of previous interpretations. **We have attempted to clarify. Our interpretations accurately delineate the fault from SAR and photography. Validating the SAR images using the photography allows us to be certain of the location and extent of the fault plane. Previously published SAR images have issues with co-eruptive plumes obscuring the view for robust interpretations to be completed. Thus our (mostly) unobscured images with accompanying photographs remove the ambiguity.**

Line 123: It is "new" for this study, or published by others? **This is the Gouhier and Paris (2019) DEM, which is new in the sense of it was published in 2019. Apologies, this is corrected for proper representation.**

Line 194: Why featureless? In line 196 you support this statement saying that a seismic line across the pre-collapse caldera shows a flat sea floor. I think that if a seismic line shows a flat sea floor you can not assume the sea floor is flat also outside the line track. It seems to me that the sea floor is featureless because of the low-resolution data. The block visible in figure 4B (line SS11-C) is not visible on the 1990 MBES because the lack of resolution. From the size of this block it seems also that the resolution of the MBES is much less than 100 m. **The block in SS11-C is indeed in the 1990 MBES, shown in the gridded data in this paper (feature 2 in figure 4) and in the Deplus et al. (1995). With regards to the “featureless” seafloor, we concede that the lower 1990 MBES resolution may preclude an absolute featureless seafloor. The seismic reflection profiles from 2017 support a the attribution of featureless as the Sparker method here, has a 4-6 m vertical resolution and there are no positive features in the basin or on the Anak Krakatau slopes (at least on the track lines). For the pre-event data, away from the track lines there no features detected, although at the lower resolution. However, there are certainly not large landslide blocks on the SW slope of Anak Krakatau in the pre-event data.**

Lines 194-196: It would be good if you can point such features with arrows. **These features are shown with arrows in the bathymetry in figure 4.**

Line 201: How do you know that the deposits accumulated between 1883 and 2017? Provide more information in support of this statement. **The caldera formed in 1883 with the caldera collapse eruption at Krakatau. The pre-event seismic reflection survey was conducted in 2017. The deposits were accumulated between these two ages. With the data presented the date range for the accumulations cannot be narrowed further.**

Line 202: “<5 m thick”, however in the methods you say “The vertical resolution is less than 10 m”. It would be good the dominant frequency at such depths so to calculate the vertical resolution, as this information is not provided in the Methods. **The vertical resolution is 4-6 m, so features at or less than 5 m can be resolved to an minimum to 4 m thickness before resolution is lost.**

Line 314: How do you know there is negligible to no drape? Do you have a sediment core? **Sediment coring failed. The currents in the straits were too hazardous. We deployed a corer to water depths of 250 m but paid-out over 400 m of cable. There is negligible drape determined from the seismic reflection profiles that show no reflectors over the landslide blocks e.g. SKC-03. However, there are instances of drape, such as where there is a single reflector over the landslide blocks e.g. SKC-02; or where there are discrete packages of draping sediments. However, generally there is negligible calculable drape.**

Line 928: “This has been previously incorrectly interpreted as”... by who? And also, do you need this sentence in a figure caption? I think many figure captions present too much "interpretation" that is not needed. **We have removed over-interpretation from the captions.**

I hope the authors will find my comments helpful.

Reviewer #3 (Remarks to the Author):

Review of "Submarine observations show half of the island of Anak Krakatau failed on December 22nd 2018" – submitted to Nature Communications

This paper presents the result of a comprehensive survey of the geological and topographical changes due to the 2018 Anak Krakatoa landslide and eruption, combining multiple different sources of information, from SAR to results to a submarine investigation, among others. A comparison between the pre- and post-event situations is given emphasis. The main new contribution is the result from a comprehensive submarine field investigation, and the interpretation of the landslide volume, deposit, and composition.

There is no doubt that this is not only a comprehensive piece of work from the data collection point of view, it is also a thorough analysis attempting to combine the various datasets to interpret the release volume, deposits, and deformation as detailed as the data allow for. However, there is no attempt to

model neither the flank collapse and the pyroclastic flows, nor the subsequent tsunami generation. Hence, the present paper mainly provides the relevant background and data interpretation that can allow for more accurate landslide and tsunami hindcasts of the present event in the future. **Modelling these processes and the addition of that information, here, would be a tremendous undertaking and would not fit within the scope of a single Nature Communications article in our opinion. Accurately understanding the parameterisation of the landslide, and providing information for tsunami hindcasting is still very important.**

Many moderate and minor aspects must be clarified and improved before a publication can be considered. Some general comments are discussed first, followed by a series of line-by-line comments (partly elaborating and overlapping with the general comments).

The first general comment concerns the interpretation of the volume uncertainty. While there is no doubt that the authors' description of the data collection and comparison with previous datasets is trustworthy and thorough, there are elements of uncertainty in the interpretations that need better attention. In particular, there seems to be uncertainties (that are not necessarily quantifiable) related to:

- Less detailed mapping of the landslide near the volcano. Seismic lines as well as pre- and post-surveys are absent or more uncertainty is present here.
- Differences in pre-event bathymetries.
- Growth and changes in the volcano between the event and the survey.
- Sedimentation and debris transported away from the study area during and after the event.
- Erosion of the antecedent material.

All these items are discussed by the authors. However, in the quantitative analysis a rather narrow uncertainty bound is presented with little discussion about possible limitations. In particular, the first three bullet points represent sources of uncertainty that are not easily quantifiable. In fact, related to the third bullet point, it is highlighted by the authors in the abstract that "...shows how extremely rapid rebuilding can obscure evidence for such collapses". Moreover, the uncertainty in the slip plane configuration (figure 8C) shows that an essential piece of the submerged volume cannot be estimated by the data. This represents a major source of uncertainty that is not sufficiently discussed. I would suggest that when presenting the volume estimates and in particular the error estimates, the authors provide a more modest approach, and discuss these sources of uncertainty more elaborately. They should consider revising the description of the quantitative error estimates, or alternatively, skip them altogether. In summary, the quantitative analysis seems overly confident, and I suggest that the authors put more focus on describing uncertainties. This might also elucidate what the given error estimates really represent. It is necessary that this discussion is given in the main body of the paper, but the authors should consider also making more active use of the supplementary material for the quantitative volume estimates. **We thank the reviewer for their thorough appraisal of uncertainty and our coverage of the topic. We were overly optimistic without other uncertainty previously, we have recalculated more conservative bounds. However, there are aspects of uncertainty that cannot be quantified and estimations of these uncertainties would yield incorrect or grossly large error bounds. This, as correctly indicated, includes resolving the bathymetry on the proximal slope and the lack of delineation of a submarine failure plane in either bathymetry or seismic reflection profiles. We feel that a thorough discussion of the uncertainty is definitely needed, but we have included information in table 1 and the methods for greater clarity. While we do not agree to removing these discussions, we do feel that they are perhaps not needed in the main body of text.**

Related to the above comment, the authors describe several places relatively fine-grained features of scales 5-10 m (e.g. blocky structures). It is hard to convince oneself from the figures (for instance, Figure 6 & 7) that such small features (erosion, blocks) can be interpreted. To this end: What is the smallest spatial resolution in the seismic interpretation? How are these features interpreted? What is the basis for the interpretation? Consider also adding direct links to the supplement on how this interpretation is carried out. The authors should also consider providing more figures with close-ups to display the blocks better for the reader. Again, there must be more emphasis on the uncertainty in the interpretations. **We have added magnifications of the landslide blocks, and specifically the internal fabric of the debris flow. The vertical resolution of the seismic reflection survey was greater than our conservative 10 m estimation, and more closer to 4-6 m. This is still a conservative estimate.**

Several relevant publications are not cited in the paper. In the line-by-line comments, some relevant additional papers are discussed. Some of them are more relevant than the ones listed by the authors to put the problem into context (e.g. Ramalho et al. and Paris et al., see below) and I was surprised not to see them cited if the intention is to understand better tsunami generation from volcano flank collapses.

Line-by-line comments

Line 39 - Is the 0.3 km³ in addition to the landslide volume? Can you say: "additional 0.3 km³ of tephra"? **Added.**

Line 51 – add "arguably" before "best-observed" **Added.**

Line 58 – It must be stressed that this paper, while thoroughly interpreting the geological footprint, does not actually attempt to reconstruct the dynamics of the event. Please add a sentence explaining this. **We have reworked the discussions sections e.g. 448-477. We have compared the deposit morphology and characteristics to determine aspects of the emplacement of the landslide and inferences about its translation downslope. We have reworded the introduction with, “understanding of failure processes and their implications in these settings.”**

Line 78-79: There is a larger list of volcanic induced tsunamis for Indonesia (e.g. Ruang, Tambora tsunamis, possibly others), please review and cite Løvholt et al. (2012), Paris et al. (2014). **Our understanding is that Ruang is more complicated. The examples provided are enough to show the point that these events are not uncommon in history and have been source of loss of life.**

Line 102-103: Please revise the sentence starting with "This will provide...". Although this paper provides a rigorous mapping of deposits, failure volumes, etc., the landslide dynamics and the tsunami generation are neither analysed nor modelled.

Line 174: This is unclear. Which previous volume interpretations are you referring to? Please provide a reference and elaborate. **We have removed this line of text as reviewers have highlighted a tone issue referring to incorrect volume estimates from other works.**

Line 179: The possible errors (uncertainties) provided throughout the document are very small. While the authors stress that interpretations are difficult due to dynamic changes in the volcanic growth, how can these volumes be interpreted with such confidence? I suggest that the authors discuss if there are other sources of uncertainty that may increase these uncertainty bounds. This is probably most relevant for the submerged part of the volume where uncertainties are presumably larger. **We have calculated uncertainties based on constraints of the equipment and assumptions within data conversions.**

Line 235: How is the erosion quantified? The scales of 5-10 m are very small compared to the resolution provided in the figures. I cannot see from the seismic signals any evidence for such an accurate estimate of erosion. Please elaborate and illustrate how this can be interpreted in more detail. If this cannot be done, the description and conclusion should be revised as it might indicate a larger uncertainty related to erosion. **Resolution of the seismic profiles was better than the conservative 10 m previously stated. The erosive depths were extracted along the seismic profiles and then interpolated across the landslide area using kriging.**

Line 247: This is related to the above comment: It is difficult to distinguish the 5-10 m size blocks with the present resolution (that this seems to be the limit of the seismic signal resolution). How were these rather small structures interpreted? **Most of the buried debris flow comprised discontinuous reflections and chaotic fabrics. There are blocks of rotated parallel reflections. Magnified regions of the debris flow are provided in the supplementary materials.**

Line 302: You may also add that the geometry altogether, i.e. the distribution of landslide material and the run-out distance, is as important as the depth of the failure for interpreting the landslide dynamics. **We have reworked much of this section. This is a good point.**

Line 309: Elaborate and explain what the presented uncertainty bound is linked to (and equally important, what is not covered in the quantification). **We have looked again at the uncertainty, and been more conservative with our error calculations. The determination of the uncertainty in the volume calculations is better covered in the methods and table 1.**

Line 319: How are the backfill sediments interpreted? Please elaborate. **There is text in lines 259-260. The sediments onlap on the blocks and the slope.**

Line 344: How is this short 5 m scale resolved in the interpretation of the seismic signal? See also above comment, as this issue appears a few places. **The resolution of the seismic survey is better than the conservative 10 m previously cited and is < 5 m.**

Line 364: Add "buried" before "debris flow". **This debris flow is clarified as the "buried" debris flow.**

Line 402: Reading this sentence you get the impression that the volume is a surprise, although several modelling studies have used this volume (and larger) in their analysis already. The estimate of the failure plane and volume even long before the event happened (Giachetti et al. 2012) was surprisingly close to the volume of the event. I suggest explaining this to set the work better into the present research context. **Much of this section has been reworked.**

Line 440: Several other studies are now published and should be discussed. They have a broader investigation of the sensitivity of the landslide configuration and dynamics. Some of these are already cited by the authors, but not discussed here. Additional modelling work include Borrero et al. (2020), Ren et al. (2020), Zengaffinen et al. (2020). These analyses highlight new elements that the authors might want to discuss. **These works are cited.**

Line 454: What type of investigation to identify pre- and post-event data is conducted close to the landslide area / shoreline to estimate the landslide volume and identify deposits close to or on top of the failure plane on the rock slope? Is there an area close to the shoreline that is not covered, or more rudimentary covered? Could you please elaborate if there are data gaps or uncertainties related to the continuous slope changes due to the volcanic growth? Moreover, there are clear differences between the 1990 and 2017 datasets. How are the uncertainties in the interpretations taking these differences into account? The readers should be given more awareness about these uncertainties. **We have stated there is a data gap in line 267. We have elaborated more concerning the differences between the datasets and the uncertainties. There is an additional table 1 with information.**

Line 461: Please rephrase "broader understanding of landslide dynamics and tsunami generation" to "a much more complete dataset and an interpretation that illuminate the emplacement process" or similar. The landslide dynamics and tsunami generation are after all not studied. **Much of the discussions have been completely reworded. This should meet the request of the reviewer.**

Line 466: Two essential works (Ramalho et al., 2015; Paris et al., 2017), where both tsunami deposit interpretation and joint modelling have been conducted, are not cited and discussed. They should be though, and they are the more relevant compared to the cited references if the authors want to discuss tsunami generation. **We have not chosen to enter a large discussion on tsunami generation. We simply state that using simple granular models and a range of proposed volumes, which our study shows demonstrate reflect a probably range, produce results similar to the observed tsunami inundation. A more thorough discussion on tsunami generation is perhaps outside the remit here.**

Line 477: Landslide modelling has evolved much since the developments in the 90's and there are now more advanced models available. More advanced granular models include $\mu(I)$ rheologies or other granular models, see e.g. Jop et al. (2006), Savage et al. (2014), or Si et al. (2018), to name some. These are more updated references valid for discussing future possibilities and should replace 11,35. However, I agree with the authors that the full complexity of the failure process cannot be captured by present models anyhow. **Savage et al. and Si et al. works have been added. The point that the full complexity cannot be captured in models has been removed as it is not wholly pertinent to the study.**

Line 490: Please stress that you are talking about the buried debris flow, not to be confused with the main landslide body. **This should have been iterated. See lines 477-479.**

Line 524-526: This may partly contradict the relatively narrow uncertainty bounds for the landslide volume. The authors should clarify better how the rapid changes in volcanic growth have been used in the interpretations. **We have provided information in text, in a table and in the methods. The post-event deposition has been taken into account for calculating the landslide volume.**

Line 588: A. Paris is the first author in this paper (not R. Paris). **Corrected.**

Line 973: Please include a legend that explains the colours of the different interpreted horizons such as the 1990 projected depth, etc. **Conscious that a legend would clutter the image. The caption should include the pertinent information.**

References

Borrero, J.C., Solihuddin, T., Fritz, H.M. et al. (2020). Field Survey and Numerical Modelling of the December 22, 2018 Anak Krakatau Tsunami. *Pure Appl. Geophys.* <https://doi.org/10.1007/s00024-020-02515-y> **Added**

Jop, P., Forterre, Y., & Pouliquen, O. (2006). A constitutive law for dense granular flows. *Nature*, 441(7094), 727-730. **Other citations sufficed, but can be added if essential.**

Løvholt, F., Kühn, D., Bungum, H., Harbitz, C. B., & Glimsdal, S. (2012). Historical tsunamis and present tsunami hazard in eastern Indonesia and the southern Philippines. *Journal of Geophysical Research: Solid Earth*, 117(B9). **Other citations sufficed, but can be added if essential.**

Paris, R., Switzer, A. D., Belousova, M., Belousov, A., Ontowirjo, B., Whelley, P. L., & Ulvrova, M. (2014). Volcanic tsunami: a review of source mechanisms, past events and hazards in Southeast Asia (Indonesia, Philippines, Papua New Guinea). *Natural Hazards*, 70(1), 447-470. **Added**

Paris, R., Bravo, J. J. C., González, M. E. M., Kelfoun, K., & Nauret, F. (2017). Explosive eruption, flank collapse and megatsunami at Tenerife ca. 170 ka. *Nature Communications*, 8, 15246. **Can be added, but felt the other citations sufficed.**

Ramalho, R. S., Winckler, G., Madeira, J., Helffrich, G. R., Hipólito, A., Quartau, R., ... & Schaefer, J. M. (2015). Hazard potential of volcanic flank collapses raised by new megatsunami evidence. *Science advances*, 1(9), e1500456. **Not added but other citations sufficed, and had to be mindful of total citations.**

Ren, Z., Wang, Y., Wang, P., Hou, J., Gao, Y., & Zhao, L. (2020). Numerical study of the triggering mechanism of the 2018 Anak Krakatau tsunami: eruption or collapsed landslide? *Natural Hazards*, 102(1), 1-13. **Added**

S.B. Savage, M.H. Babaei, T. Dabros (2014), Modeling gravitational collapse of rectangular granular piles in air and water, *Mechanics Research Communications* 56, 1-10, <https://doi.org/10.1016/j.mechrescom.2013.11.001>.

Si, P., Shi, H., & Yu, X. (2018). Development of a mathematical model for submarine granular flows. *Physics of Fluids*, 30(8), 083302. **Added**

Zengaffinen, T., Løvholt, F., Pedersen, G. K., & Muhari, A. (2020). Modelling 2018 Anak Krakatoa flank collapse and tsunami—effect of landslide failure mechanism and dynamics on tsunami generation. *Pure Appl. Geophys* <https://link.springer.com/article/10.1007/s00024-020-02489-x> **Added**

REVIEWER COMMENTS

Reviewer #1 (Remarks to the Author):

A second time i was reading the manuscript and found it has much improved! The authors considered almost all points raised by me and could convincingly argue those points that were beyond the scope of the manuscript or below resolution of data (e.g. in photographs). Therefore i am inclined to recommend to go ahead with publication.

Only, i realized that some questions in the reviewer letters were not answered, and some suggestions and changes requested on improving the figures were not made.

Still the print out figures are in places unreadable. If authors do not change it, it is unpublishable. See my earlier comments. Authors can also simply print out the figures and ask their office neighbors if they can decipher all. I can't read details in fig 1c, 1d, and some others. In some figures, there seem typos or misplaced text boxes, e.g. in fig 5b a "fault" text seems misplaced. I suggest let the authors to carefully check once more all the figures, and then go ahead with publication of this very important contribution.

Reviewer #3 (Remarks to the Author):

Review of "Submarine landslide megablocks show half of Anak Krakatau failed on December 22nd 2018" – revised manuscript to Nature Communications

General comments:

The revised manuscript and the rebuttal letter indicate that the majority of my comments to the first version of the paper are satisfactory embedded in the new version of the manuscript, although there is still a small list of amendments that would be necessary to be incorporated before the manuscript could be considered for publication.

I appreciate that a more in-depth discussion about uncertainties, and the new table listing the different uncertainties are welcomed. However, it is clear from the rebuttal letter that the authors are aware that not all uncertainties can be quantified, and that bounds for such uncertainties could be excessive, or as the authors write "there are aspects of uncertainty that cannot be quantified and estimations of these uncertainties would yield incorrect or grossly large error bars". This is an important and honest statement, but a statement of this kind is still absent in the main body of the paper. Importantly, it needs to be conveyed to the readers and not only to this reviewer. It is an important premise for modellers and to be aware of. Such a remark should be put into the main body of the paper, for instance in the conclusion or discussion section where this issue is considered.

Closely linked to the lack of ability to uncertainty quantification for certain features, is the issue that there are different hypotheses in the paper on shallow vs deep failure. The differences between these are larger than the quantified uncertainty ranges. Comparing the drawing of these failure planes the first version of the manuscript to the revised one, the roughly sketched failure plane area has largely changed. It should be noted clearly that these are not incorporated in the volume uncertainty.

The way of presenting different volumes, including the chronology unravelling the volumes with initial estimates etc, are scattered across the text and hence becomes difficult to follow at places. It could be worthwhile including a table where these different volumes are compiled, along with a list of brief explanations. For instance: in the abstract the authors use 0.236 km³ (is this correct by the way?) while in the conclusions 0.21 km³ is used. Then, on line 387, you present the total post eruptive volume of 0.299 km³. I think it would strengthen the paper if the readers would have a simpler access to a list of these volumes and what they represent, for instance

distinguishing which part of this volume range that contributed to the tsunami generation, and which part did not.

Previous comment not yet adequately implemented:

Previous comment Line 466: Two essential works (Ramalho et al., 2015; Paris et al., 2017), where both tsunami deposit interpretation and joint modelling have been conducted, are not cited and discussed. They should be though, and they are the more relevant compared to the cited references if the authors want to discuss tsunami generation.

Authors rebuttal: We have not chosen to enter a large discussion on tsunami generation. We simply state that using simple granular models and a range of proposed volumes, which our study shows demonstrate reflect a probably range, produce results similar to the observed tsunami inundation. A more thorough discussion on tsunami generation is perhaps outside the remit here.

Reviewer Response: The authors are missing the point, these references (re-emphasised below) are necessary. The entire motivation for this paper is understanding volcano flank tsunamis and emplacement mechanisms, so stating that this is not needed is a big contradiction. In the first sentence of the paper, the authors are citing four works (1-4) and state that "the factors governing volcanic edifice instability and landslide-emplacement remain incompletely understood." The two works cited above are unique because they bring together submarine geology, paleotsunami observations, through modelling. The point is that the tsunami observations will provide much more information than the static field landslide run-out deposit itself, and by bringing this together, deeper insight can be revealed. Paris et al and Ramalho et al did just that. They are also more updated and bring more information to the table than refs 1-4 with this respect, and hence needs to be cited. If there are still too many references I rather suggest some of 1-4 to be removed.

Line-by-line comments:

Line 33 – It is unclear to me if the 0.236 km³ is the volume you want to convey here.

Line 36 – Add "several" after "resolving". All ambiguities related to this complex event are not yet resolved.

Line 45 – As already discussed, please add the two references given below.

Line 86 – Add "landslide and" after "numerical". After all it is the landslide model you match to the observations.

Lines 282-285 – As said above, this issue must be leveraged a more up front or in the discussion and conclusion, indicating that the non-quantifiable uncertainties related to the different failure plane hypotheses can be larger than the volumetric uncertainty bounds.

Lines 387-389 – Please delete this statement, as it is misleading. It was also indicated by other reviewers previously to tone down statements of this kind.

Lines 410-412 – The authors could perhaps add here rough numbers of what the volumetric range between these two different volumetric interpretations would be?

Line 426 – Not all studies are listed in the table, the table should be completed to add all the numerical studies cited in the paper. Otherwise the table could potentially give a bias picture.

Lines 452-454 – Landslide modelling studies cited would give estimates of these velocities that are probably better constrained the figures given in ref 5.

References:

Paris, R., Bravo, J. J. C., González, M. E. M., Kelfoun, K., & Nauret, F. (2017). Explosive eruption, flank collapse and megatsunami at Tenerife ca. 170 ka. *Nature Communications*, 8, 15246.
Ramalho, R. S., Winckler, G., Madeira, J., Helffrich, G. R., Hipólito, A., Quartau, R., ... & Schaefer, J. M. (2015). Hazard potential of volcanic flank collapses raised by new megatsunami evidence. *Science advances*, 1(9), e1500456.

**National
Oceanography Centre**

NATURAL ENVIRONMENT RESEARCH COUNCIL

National Oceanography Centre
European Way
Southampton SO14 3ZH
United Kingdom

Tel: +44 (0)23 8059 6666

<http://noc.ac.uk>

Direct line: 02380 596577

Email: James.Hunt@noc.ac.uk

23 December 2020

Re: resubmission of article NCOMMS-20-17603B – response to reviewer comments

Dear Reviewers,

Please find our responses to reviewer comments below.

We look forward to hearing your response,

Yours sincerely,

Dr James Hunt
Senior Researcher
Marine Geoscience Department
National Oceanography Centre, UK

Author comments in **bold**. Reviewer responses to address are highlighted in *italics*.

Reviewer #1 (Remarks to the Author)

A second time I was reading the manuscript and found it has much improved! The authors considered almost all points raised by me and could convincingly argue those points that were beyond the scope of the manuscript or below resolution of data (e.g. in photographs). Therefore I am inclined to recommend to go ahead with publication.

Only, I realized that some questions in the reviewer letters were not answered, and some suggestions and changes requested on improving the figures were not made.

Still the printout figures are in places unreadable. If authors do not change it, it is unpublishable. See my earlier comments. Authors can also simply print out the figures and ask their office neighbors if they can decipher all. I can't read details in fig 1c, 1d, and some others. In some figures, there seem typos or misplaced text boxes, e.g. in fig 5b a "fault" text seems misplaced. I suggest let the authors to carefully check once more all the figures, and then go ahead with publication of this very important contribution.

New rebuttal: Apologies. The figures were adapted, and text was enlarged. Home-printing has been low quality and the readable text was attributed to that rather than faults in the new figures. This did not allow appropriate quality control. The figures, and labelling in particular, have all been thoroughly checked and where necessary have been reformatted to ensure that they are legible at print scale.

Reviewer #3 (Remarks to the Author based on Reviewer #2 past comments)

Initial reviewer comment: The paper from Hunt et al. reports on the collapse of the Anak Krakatau volcanic Island, which occurred on December 22nd 2018.

The authors integrate satellite, bathymetric, and seismic data to quantify the total sediment volume that collapsed into the sea, providing for the first time a quantitative estimate of the subaerial and submarine components of the collapse. Compared to many other studies that investigated the 2018 Krakatau eruption, this is the first work presenting multibeam bathymetric data acquired soon after the event (which I think is the most relevant dataset) and higher resolution satellite images. Thanks to the new bathymetric data, the authors were able to quantify a total landslide volume of $0.214 \pm 0.025 \text{ km}^3$, of which $0.098 \pm 0.005 \text{ km}^3$ related to the subaerial failure. These values are overall similar to the results of previous studies, as Grilli et al. (2019) quantified a primary landslide volume in the range of $0.22\text{--}0.30 \text{ km}^3$, and Walter et al. (2019) and Gouhier and Paris (2019) quantified a subaerial failure volume of 0.102 km^3 and 0.938 km^3 , respectively. Other studies, as Williams et al. (2019) proposed a much smaller total volume. The authors interestingly highlight that the post-failure deposition during the regrowth of the volcano may obscure both the subaerial and submarine scars, thus affecting quantification of the total collapsed volume. The authors then present two possible scenarios for the failure geometry (with two failure planes at different depths), of which model 2 looks very similar to the model presented in Grilli et al. (2019). The discussion starts with a comparison between the failed volume calculated in this study and the ones calculated in previous works. This comparison mainly points to the fact that the failure geometry of model 1, with a failure plane that cuts the seafloor at around -120 m water depth, is the most reasonable interpretation. Despite this conclusion can help a better understanding of the Krakatau event, it is not clear what this conclusion means in terms of a deeper understanding of the collapse processes that may occur in volcanic islands. The second part of the discussion mainly focuses on the morphology of the submarine landslides, and the authors suggest that the presence of blocks indicate a purely translational transport mechanism of relatively coherent masses downslope. Similar features have been observed in many other settings, modern and ancient, and interpreted in a similar way.

Rebuttal: The Anak Krakatau event has a known measured tsunami, thus not only does the discovery of a translational slide in this setting important, but it can be tied to the generation of this particular tsunami. However, Ritter experienced a flank collapse of a similar relative scale, but despite a similar composition the failure has highly disintegrative and erosive. Here, this translational slide only travelled 1.5 km into the basin, and maintained relatively intact. In comparison, the Ritter collapse generated an over 70 km run-out failure and debris flow.

Response: The differences between Ritter and Anak Krakatau (AK) are acknowledged. It is clear that the thorough documentation and reconstruction of the event, and the wealth of data available, have clear value both due to the elaboration of this investigation and because the event is recent. The point the authors refer to on translational sliding needs a remark though. As this is a subaerial event, for which tsunami generation is governed by the impulsive motion of the slide hitting the water. Tying this to tsunami generation can therefore be misleading. Moreover, there are many subaerial slides involving this mechanism. It is hence not unique just because Ritter is different from AK.

New rebuttal: In responding to the initial reviewer comment, we feel that it is worth reiterating the importance and novelty of completing such a detailed subaerial and submarine survey of a volcanic island flank collapse so quickly after the event. The post-event volcanism and island reconstruction show that these environments are very dynamic, and that much of the important details of these slide studies can be quickly lost.

Towards the references to Anak Krakatau vs Ritter, despite similarities in island construction and volcanism, and in the nature of the failure plane (in both cases involving substantial subaerial and submarine components, cutting the

central volcanic conduit beneath sea level), the landslide deposits demonstrate that the fragmentation and transport mechanisms at Ritter and Anak Krakatau were, somewhat surprisingly, very different. Anak Krakatau predominantly involved translational large block motion with minor deformation and Ritter experienced a highly disintegrative debris flow and associated turbidity current. We refer to Ritter not only because of the potential dichotomy but because it is a relatively recent historic event, and with such events having low frequencies historically, it is important to compare the gaps in available data between the 1888 Ritter and 2018 Anak Krakatau events. We agree that the translational nature of block motion at Anak isn't key to tsunami generation, but the point we are making is that, *despite* these differences in landslide fragmentation and breakup, both events were still very efficiently tsunamigenic. We are therefore in agreement with the reviewer, that it is the initial phase of motion that is most important for tsunami generation, but we feel the differences in the nature of the deposits are important to point out, because for nearly all past events, the deposit is the primary source of data and interpretation of the parent landslide. We continue the comparison between Anak Krakatau and Ritter in the discussion because we feel the dichotomy between the two events, potentially representing end members of landslide emplacement processes for collapses at rapidly constructed cones, is important to highlight. While there are examples of other potential slides that could offer similarities to Anak Krakatau and Ritter, most are not historical, lack detailed subaerial data and submarine data, lack valid pre-event reconstructions and lack documented evidence of the event at the time.

Although tsunamigenesis is primarily affected by the impulsive motion of the failure mass displacing water, but the rheology of the materials, fragmentation of the mass and nature of the slide transport does impart influence on the wave(s) generated, and this is a further reason to highlight the nature of the Anak Krakatau deposits. Thus, parameterization of such processes and features is warranted. Both reviewers allude to the fact that there are additional settings, modern and ancient, with deposit features interpreted similarly to Anak Krakatau. However, Anak Krakatau represents the most recent and best observed event. While processes at Anak Krakatau may have been interpreted elsewhere, the surveys at Anak Krakatau, so soon after event, provide greater validity towards conclusions drawn from the data, and novelty to the results.

Initial reviewer comment: Also this second part of the discussion does not provide a clear step forward in knowledge, apart from better describing the specific case study. The third part of the discussion focuses on the preconditioning factors, and the authors concluded that loading of the SW flank by lavas and volcanoclastic sediment coupled with topographic deformation preconditioned the failure. Walter et al. (2019) presented a similar conclusion.

Rebuttal: We now show additional information that show additional preconditioning factors. We show the brittle deformation and occurrence of surface fissures that support Walter et al. suggestion of SW deformation of the flank. We also show evidence of fluid flow in the form of fumarole activity showing active venting over the region that would be affected by the final failure.

Response: Can this be of general importance for tsunami generation? In response to the reviewer statement below, this is still not clear. To my understanding this is still first and foremost important for understanding this scenario, more general statements (e.g. lines 57-58) should be revised and focus more on the AK event. The formulation in lines 506-508 appears more balanced.

New rebuttal: We have reframed the lines in the introduction in-line with the more balanced concluding lines. While we want to be able to provide more detailed and accurate information to inform tsunami models for this event, it is also a desire to better understand the landslide and emplacement mechanism in general. It is hoped that by re-orientating focus on the Anak Krakatau event, we have not removed some of generality that could be translated to offer scenarios, given that a more general perspective was requested by both reviewers and editor. We focus attention on better resolving the Anak Krakatau event, while stipulating that by better resolving this event both our understanding of other similar events and modelling their tsunamis will also be better informed. We hope this strikes a more balanced approach for the introduction in lines 57-60.

Initial reviewer comment: In summary, I found the discussion very descriptive, and the key statement presented in the introduction ("detailed reconstructions are key to advancing our broader understanding of tsunami generation from volcanic island flank collapses") is not further developed here. Overall, the manuscript is well written, however, some parts can be improved. There are different statements that, despite logical, should be supported by additional data, as direct sediment sampling of the seabed. See my comments below. The figures are clear, sometimes probably too dense. Some of the figures present errors and typos, one in particular (Figure 8) looks like a screen snapshot as you can see words highlighted by a red line and a selection box for the text. The captions also require some work, as information is missing. See my comments below. An additional important point is the public availability of the data presented here. I did not find a statement on data availability, so I refer this concern back to the editor.

Rebuttal: We now show additional information that show additional preconditioning factors. We show the brittle deformation and occurrence of surface fissures that support Walter et al. suggestion of SW deformation of the flank. We also show evidence of fluid flow in the form of fumarole activity showing active venting over the region that would be affected by the final failure.

Response: The publisher have to decide if this is sufficient, at least the issue is pointed out.

NEW rebuttal: The additional details from our study comport and expand the findings of Walter et al towards preconditioning of the volcanic flank for failure. These additional details are not the focus of our study, but do provide new novel information on preconditioning, both supporting previous interpretations and providing nuance in the detail of the failure. The new information is also important in providing some further constraints on the failure models, as discussed later. With regards to additional data, direct sampling was not possible due to the vessel limitations and currents, and most importantly the fact that the landslide is buried beyond the reach of conventional gravity/piston coring.

Initial reviewer comment: Considering the points above, I do not think this study introduces a major step forward in knowledge that is required by Nature Communications. This study presents a better quantification of the Krakatau event, but not a deeper understanding of the processes and products related to the collapse of volcanic islands that can be applied elsewhere. I think the authors does not present, and test, any new hypothesis that can help understanding such phenomena and evaluating the associated risk. For example: Why some volcanic collapses generate coherent blocks that move downslope for few kilometers, while others, as the 1888 Ritter Island event, resulted in a debris flow moving tens of kilometers away?

Rebuttal: This comparison to Ritter is valid, and highlights the importance of the study. A more disintegrative and long run-out slide was expected here, at Anak Krakatau. Another key aspect that does show a step change is the huge volume of post-event materials produced, and that the majority of these volcanoclastic sediments are deposited offshore. Thus deposit the rapid recovery of the island post-event, this rapid recovery actually belies the true scale of recovery and the amount of volcanic debris produced. However, the major novel feature of this study is the combination of subaerial and submarine investigations of the slide so soon after the event. We have outlined the novelty and importance of our study as requested. Our study documents, for the first time, direct and complete high-resolution observations of the landslide deposits from the Anak Krakatau flank collapse on December 22nd, 2018. Firstly, there is the novelty of the completion of the submarine studies of this contemporary volcanic island flank collapse so soon after event, at very high resolution, and with pre-event data for comparison. However, this is not the only novelty, but does warrant highlighting and emphasising. It is important to stress the novelty of this opportunity and the data we have brought together for this study of both the subaerial and submarine components of the flank collapse at Anak Krakatau. Secondly, the Anak Krakatau event has given us greater insights into the failure and emplacement processes in these settings. The Anak Krakatau event has shown that landslides in these settings can remain predominantly intact and translate only a limited distance into the basin. In contrast, similar destructive events at similar volcanic islands are more disintegrative, more erosive and travel further into the basin (such as Ritter, as suggested by reviewers). Thus, this study shows the dichotomy of potential slides in these volcanic settings capable of generating destructive tsunamis. Thirdly, we show the excessive additional generation of tephra from the co-eruptive events at the time of the flank collapse, and that these materials are deposited in the submarine realm on the slope and caldera basin. Single pyroclastic flows entering the submarine realm, such as at Montserrat, have been studied but have demonstrated that these deposits from extreme volcanic events are often sub-metre in thickness. Meanwhile eruptive sequences tens of metres thick often represent thousands of years of sediment accumulation. Here, at Anak Krakatau our results show an unprecedented thickness of co-eruptive and immediately post-event volcanoclastic accumulation from less explosive Surtseyan activity. We show that the new volume estimate from these eruptions is greater than the volume lost from the island flanks during the flank collapse.

Response: This shows both the relevance and the limitation of the study, which is already outlined in the previous review and the responses above: I agree about the claim of novelty, relevance, and importance related to AK. Moreover, the new experience from post event volume interpretation is clearly novel. But it is not obvious how this morphological analysis can help explaining flank collapse and more generally landslide tsunamis from subaerial slides. For the latter, it is the volume, geometry, and runout interpretation that matters due to the impulsiveness such events display, and this have been largely explored elsewhere and should be subject to another study as the authors say.

NEW rebuttal: We agree with the second reviewer comments about the novelty of our study. The majority of previous studies, including the events the reviewer highlights from the requested citations, are from landslides in the geological record, and it is important to recognize that in nearly all cases there are no constraints at all on the associated tsunami. While modelling approaches can therefore address how variable failure and emplacement processes influence wave generation, the novelty and importance of the Anak Krakatau event is its combination of direct failure and landslide observations (presented here) alongside direct tsunami measurements. Although the reviewer is correct in outlining that the morphological characteristics most importance towards understanding tsunamigenesis are volume, geometry and runout, we make the important observation here that the runout of the Anak Krakatau deposits is limited in comparison with many landslide deposits observed around volcanic islands elsewhere, and yet the event was clearly an efficient tsunami source.

Our new morphological analysis of the subaerial failure provides a more accurate assessment of the subaerial volume and the geometry of the failure surface where it breached the surface. This information is important as it indicates that the main pyroclastic cone itself was incorporated in the failure, with the overall form of the failure plane being similar to Ritter. Our analysis provides a comprehensive failure scenario, constrained by all available subaerial and submarine data, which is essential for testing and validating tsunami modelling of the event. In terms of further

general insights, we indicate and discuss implications of our data towards preconditioning of the failure (as per above), which provides a holistic parameterization of the slide event.

Initial reviewer comment: How the knowledge gained in this field area can be used to predict the style of failure (and thus the risk to submarine infrastructure or the tsunamigenic potential) of volcanic islands in different settings?

Rebuttal: The study shows the flaws in predicting the failure extent and style, especially without all the information. For example, the information of fissures and fumarole venting in this study show the SW flank deformation in response to loading from the lavas and Strombolian ejecta, while the venting occurred along the E and S extent of the failure. Knowledge of the past slides, as shown here, also defined a consistent weakness defining the N extent of the failure. Thus with this information, it may have been possible to predict the size of the subaerial failure.

Response: This does not really address the issue, which is the emplacement mechanism during tsunami generation. This necessitates numerical modelling, which is not done here.

New rebuttal: Agreed, the rebuttal did not completely address the comment. Indeed, resolution of the failure and emplacement mechanisms during tsunami generation is important. We agree that understanding a failure and emplacement mechanism would be assisted by numerical modelling of the landslide, but this would also only be valid with a detailed understanding of the internal structure of the edifice and its mechanical properties. This is an important aim of future research, but is not in the scope of this study. A full understanding will require direct study of the failure material and more detailed analyses of deformation in the pre-collapse period, alongside analysis of accompanying volcanic processes. These datasets do not yet exist, but this doesn't detract from the merit of the results presented here, which provide important constraints on the fragmentation and transport of the failure mass, its volume (and hence the position of the submarine failure plane) and on the subaerial limits of failure. It is the timely nature of the survey that has uniquely allowed us to identify these aspects of the failure. While we outline a potential dichotomy to two end-member slide examples (translational vs disintegrative), we do also state that predicting the ultimate style of failure from any particular volcanic island flank would be very difficult ahead of the event. Thus, we state we cannot accurately predict the style of failure, but highlight that the failure may lie between two possible end-member types and that a great deal of proprietary information would be needed to make any further assessment of the likely failure type to occur in the future.

Initial reviewer comment: How such detailed calculation of the landslide volume affects the results of tsunami modelling?

Rebuttal: Producing tsunami models is a feature of follow on work. Completing the work here would make the study excessively long and not suitable for this format. The tsunami modelling warrants a focused study.

Reviewer comment: I think the authors should make an effort in improving the discussion, stating (and testing) one or more new hypotheses that will help understanding such kind of phenomena.

Rebuttal: We have reframed the paper to highlight the novel science being presented.

Response: Thank you for improving the discussion part, with the reorganisation this reads in a more balanced way now.

NEW rebuttal: Input from the reviewers has definitely allowed completion of a more structured manuscript and we thank the reviewers for their comments

Additional Comments:

The additional comments of reviewer 2 were not further commented on. They were addressed in the previous manuscript draft.

Reviewer #3 (Remarks to the Author):

Review of "Submarine landslide megablocks show half of Anak Krakatau failed on December 22nd 2018" – revised manuscript to Nature Communications

General comments:

The revised manuscript and the rebuttal letter indicate that the majority of my comments to the first version of the paper are satisfactory embedded in the new version of the manuscript, although there is still a small list of amendments that would be necessary to incorporate before the manuscript could be considered for publication.

I appreciate that a more in-depth discussion about uncertainties, and the new table listing the different uncertainties are welcomed. However, it is clear from the rebuttal letter that the authors are aware that not all uncertainties can be quantified, and that bounds for such uncertainties could be excessive, or as the authors write "there are aspects of uncertainty that cannot be quantified and estimations of these uncertainties would yield incorrect or grossly large error bars". This is an important and honest statement, but a statement of this kind is still absent in the main body of the paper. Importantly, it needs to be conveyed to the readers and not only to this reviewer. It is an important premise for modellers and to be aware of. Such

a remark should be put into the main body of the paper, for instance in the conclusion or discussion section where this issue is considered.

NEW rebuttal: We agree. We have added the text as written to the main body of the text where we discuss uncertainties at around line 331-333.

Closely linked to the lack of ability to uncertainty quantification for certain features, is the issue is that there are different hypotheses in the paper on shallow vs deep failure. The differences between these are larger than the quantified uncertainty ranges. Comparing the drawing of these failure planes the first version of the manuscript to the revised one, the roughly sketched failure plane area has largely changed. It should be noted clearly that these are not incorporated in the volume uncertainty.

NEW rebuttal: These are updated, indeed the large failure scenario has a relatively large error, and we are open about the uncertainties. This uncertainty in the failure plane is unavoidable because of burial by post-collapse volcanic deposits, but, as we emphasise, we are able to independently evaluate the event volume from the deposit, and the submarine dataset (which is not published in any previous interpretations of the failure) is the key to this. We demonstrate that our estimated shallower failure plane is most consistent with the deposit volume.

The way of presenting different volumes, including the chronology unravelling the volumes with initial estimates etc, are scattered across the text and hence becomes difficult to follow at places. It could be worthwhile including a table where these different volumes are compiled, along with a list of brief explanations. For instance: in the abstract the authors use 0.236 km³ (is this correct by the way?) while in the conclusions 0.21 km³ is used. Then, on line 387, you present the total post eruptive volume of 0.299 km³. I think it would strengthen the paper if the readers would have a simpler access to a list of these volumes and what they represent, for instance distinguishing which part of this volume range that contributed to the tsunami generation, and which part did not.

NEW rebuttal: We accept that our discussion of volumes in the abstract and main text was dense, and could potentially cause confusion. This has been changed to ensure consistent and clear values are cited throughout. We have also added a table in the supplementary information, as suggested.

Previous comment not yet adequately implemented:

Previous comment Line 466: Two essential works (Ramalho et al., 2015; Paris et al., 2017), where both tsunami deposit interpretation and joint modelling have been conducted, are not cited and discussed. They should be though, and they are the more relevant compared to the cited references if the authors want to discuss tsunami generation.

Authors rebuttal: We have not chosen to enter a large discussion on tsunami generation. We simply state that using simple granular models and a range of proposed volumes, which our study shows demonstrate reflect a probably range, produce results similar to the observed tsunami inundation. A more thorough discussion on tsunami generation is perhaps outside the remit here.

Reviewer Response: The authors are missing the point, these references (re-emphasised below) are necessary. The entire motivation for this paper is understanding volcano flank tsunamis and emplacement mechanisms, so stating that this is not needed is a big contradiction. In the first sentence of the paper, the authors are citing four works (1-4) and state that "the factors governing volcanic edifice instability and landslide-emplacement remain incompletely understood." The two works cited above are unique because they bring together submarine geology, paleotsunami observations, through modelling. The point is that the tsunami observations will provide much more information than the static field landslide run-out deposit itself, and by bringing this together, deeper insight can be revealed. Paris et al and Ramalho et al did just that. They are also more updated and bring more information to the table than refs 1-4 with this respect, and hence needs to be cited. If there are still too many references I rather suggest some of 1-4 to be removed.

NEW rebuttal: Apologies, our comment regarding how these papers would be referred to later in the manuscript may have been misunderstood. We agree that discussion and inclusion of these papers in the earlier stages of the manuscript is appropriate and also serves to strengthen the case for the current study's importance.

Line-by-line comments:

Line 33 – It is unclear to me if the 0.236 km³ is the volume you want to convey here.

This has been updated to the corrected 0.214 km³ volume after the corrections are applied. The aforementioned volume is the pre-corrected volume, it is more accurate and correct to state the 0.214 km³ volume as advised by the reviewer.

Line 36 – Add "several" after "resolving". All ambiguities related to this complex event are not yet resolved.
Agreed, and corrected.

Line 45 – As already discussed, please add the two references given below.
These are added here. Although they are also discussed more thoroughly later in the manuscript.

Line 86 – Add "landslide and" after "numerical". After all it is the landslide model you match to the observations.
Agreed, and corrected.

Lines 282-285 – As said above, this issue must be leveraged a more up front or in the discussion and conclusion, indicating that the non-quantifiable uncertainties related to the different failure plane hypotheses can be larger than the volumetric uncertainty bounds.

Text has been added around line 423 to iterate the point outlined by the reviewer. We agree that it is needed, and should have indeed highlighted this more.

Lines 387-389 – Please delete this statement, as it is misleading. It was also indicated by other reviewers previously to tone down statements of this kind.

We understand the reviewer, however, this statement is not meant to be a negative comment towards past works, but we feel it is important to highlight both how and why our volumes differ from previous approaches, and that the availability of marine data is important in enabling more accurate volume interpretations to be drawn. We want to draw attention to the point that to make our conclusions on this eruptive volume it was important to have access to the data we collected.

Lines 410-412 – The authors could perhaps add here rough numbers of what the volumetric range between these two different volumetric interpretations would be?

These are added. The values for these models are added with rough volume ranges. Both volumes model have been modified to steepen the slope more accurately to ensure that the vent elevation is below sea level, thus the values are higher, but the results do not compromise the results or conclusions. These values are 0.175 and 0.313 km³, respectively, with error estimates of 0.015 km and 0.043 km³, respectively. This strongly favours a shallower collapse plane, as previously concluded.

Line 426 – Not all studies are listed in the table, the table should be completed to add all the numerical studies cited in the paper. Otherwise the table could potentially give a bias picture.

All the studies should now be included.

Lines 452-454 – Landslide modelling studies cited would give estimates of these velocities that are probably better constrained the figures given in ref 5.

Surprisingly few of the modelling references provide slide velocities. A range has been now been cited.

References:

Paris, R., Bravo, J. J. C., González, M. E. M., Kelfoun, K., & Nauret, F. (2017). Explosive eruption, flank collapse and megatsunami at Tenerife ca. 170 ka. Nature Communications, 8, 15246.

Added.

Ramalho, R. S., Winckler, G., Madeira, J., Helffrich, G. R., Hipólito, A., Quartau, R., ... & Schaefer, J. M. (2015). Hazard potential of volcanic flank collapses raised by new megatsunami evidence. Science advances, 1(9), e1500456.

Added and discussed.

REVIEWERS' COMMENTS

Reviewer #3 (Remarks to the Author):

Review of "Submarine landslide megablocks show half of Anak Krakatau failed on December 22nd 2018" – second revision of manuscript to Nature Communications

The paper now reads well, and the comments to the previous version of the manuscript and the related amendments were mostly adequately addressed. Reading the revised paper, only a few, mostly minor comments remain. If these comments listed below, are adequately addressed, I believe the paper could be suitable for publication:

Line 27 – The term "poorly understood" does not fully cohere with the text below, I would suggest to use the phrase "incompletely understood" or a similar statement as also used in the introduction. "Poorly understood" is probably too strong.

Line 109-110 – A slight rephrase is needed, as there are other smaller events that are similarly well documented. Instead of "the only such event" state "the only major island arc collapse".

Line 138 and 142, and conclusions: If I'm reading this correct, this is a precursor, that if treated and understood adequately at the time, could have been use for tsunami preparedness purposes. The authors may want to elaborate slightly on this.

Line 320: Table 1 is very useful, but it would be even more useful if the estimated volumes were given explicitly in addition to the uncertainties. This would help readability tremendously.

Line 331: This sentence does not read well? Please revise.

Line 339: This seems to be taken straight from rebuttal letter conversation, but in this context the last part of the sentence does not read well here and is partly contradictory. In cases where you cannot make quantification based on data, expert assessment is used. I would suggest that you delete the second part of the sentence after the comma, and rather make a statement that scientific judgement, for instance related to the failure planes, are used in these situations to come up with alternatives.

Line 545 – add "major" before "volcanic"

**National
Oceanography Centre**

NATURAL ENVIRONMENT RESEARCH COUNCIL

National Oceanography Centre
European Way
Southampton SO14 3ZH
United Kingdom

Tel: +44 (0)23 8059 6666

<http://noc.ac.uk>

Direct line: 02380 596577

Email: James.Hunt@noc.ac.uk

30 October 2020

Re: resubmission of article NCOMMS-20-17603B – response to reviewer comments

Dear Reviewers,

Please find our responses to reviewer comments below.

We look forward to hearing your response,

Yours sincerely,

Dr James Hunt
Senior Researcher
Marine Geoscience Department
National Oceanography Centre, UK

Reviewer #3 (Remarks to the Author):

Review of "Submarine landslide megablocks show half of Anak Krakatau failed on December 22nd 2018" – second revision of manuscript to Nature Communications. **Author responses in bold.**

The paper now reads well, and the comments to the previous version of the manuscript and the related amendments were mostly adequately addressed. Reading the revised paper, only a few, mostly minor comments remain. If these comments listed below, are adequately addressed, I believe the paper could be suitable for publication:

Line 27 – The term "poorly understood" does not fully cohere with the text below, I would suggest to use the phrase "incompletely understood" or a similar statement as also used in the introduction. "Poorly understood" is probably too strong. **This is corrected and edited as suggested.**

Line 109-110 – A slight rephrase is needed, as there are other smaller events that are similarly well documented. Instead of "the only such event" state "the only major island arc collapse". **This is edited as suggested.**

Line 138 and 142, and conclusions: If I'm reading this correct, this is a precursor, that if treated and understood adequately at the time, could have been use for tsunami preparedness purposes. The authors may want to elaborate slightly on this. **Text is added to the conclusions line 590. We have not gone further to specifically imply links to tsunamis preparedness. These processes could be monitored and there presence and/or increased activity could indicate the potential of a flank collapse occurring linked to volcanic activity. However, it cannot be said that if these features had been identified or understood adequately at the time then there could have been increased tsunami preparedness. Identifying these processes (especially flank deformation and fluid flow) may indicate that risk of mass wasting has increased, but cannot at this point place a timing between their occurrence and mass wasting occurring.**

Line 320: Table 1 is very useful, but it would be even more useful if the estimated volumes were given explicitly in addition to the uncertainties. This would help readability tremendously. **Estimated volumes have been added.**

Line 331: This sentence does not read well? Please revise. **This has been edited.**

Line 339: This seems to be taken straight from rebuttal letter conversation, but in this context the last part of the sentence does not read well here and is partly contradictory. In cases where you cannot make quantification based on data, expert assessment is used. I would suggest that you delete the second part of the sentence after the comma, and rather make a statement that scientific judgement, for instance related to the failure planes, are used in these situations to come up with alternatives. **This is line 349. The text has been altered as suggested and uses the example of extrapolating the trajectory of the failure plane, as suggested.**

Line 545 – add "major" before "volcanic". **This is amended as suggested.**